# KEY DESIGN CHOICES FOR DOUBLE-TRANSFER IN SOURCE-FREE UNSUPERVISED DOMAIN ADAPTATION

## ABSTRACT

Fine-tuning and Domain Adaptation emerged as effective strategies for efficiently transferring deep learning models to new target tasks. However, target domain labels are not accessible in many real-world scenarios. This led to the development of Unsupervised Domain Adaptation (UDA) methods, which only employ unlabeled target samples. Furthermore, efficiency and privacy requirements may also prevent the use of source domain data during the adaptation stage. This particularly challenging setting, known as Source-free Unsupervised Domain Adaptation (SF-UDA), is still understudied. In this paper, we systematically analyze the impact of the main design choices in SF-UDA through a large-scale empirical study on 500 models and 74 domain pairs. We identify the normalization approach, pre-training strategy, and backbone architecture as the most critical factors. Based on our observations, we propose recipes to best tackle SF-UDA scenarios. Moreover, we show that SF-UDA performs competitively also beyond standard benchmarks and backbone architectures, performing on par with UDA at a fraction of the data and computational cost. Experimental data and code will be released.

## 1 INTRODUCTION

The recent success of deep neural networks (DNNs) in many tasks and domains often relies on the availability of large annotated datasets. This can be tackled by pre-training DNNs on a large dataset and then fine-tuning their weights with target task data (Huh et al., 2016; Yosinski et al., 2014; Chu et al., 2016). Furthermore, fine-tuning is usually simpler to perform and faster than training the model from scratch, the dataset size can be smaller, and the final performance is typically higher (with some exceptions: see (Kornblith et al., 2019)). This approach is very convenient: the model requires a single expensive pre-training and can later be re-used for multiple down-stream tasks. This is a good example of transfer learning (Zhuang et al., 2021), which leverages on the information acquired from a task to improve accuracy on another task of interest.

Two relevant examples of transfer learning are Domain Adaptation (DA), that, given different–yet related–tasks, exploits source domain(s) data to improve performance on different known target domain(s), and Domain Generalization (DG), that aims to generalize to unknown target(s). As opposed to fine-tuning, in which the pre-training and downstream tasks can be significantly different, DA and DGen require stronger assumptions on the similarity between tasks, e.g., leveraging synthetic images to improve the classification of real images that share the same label space. DA is also related to Multi-task Learning (MTL) (Caruana, 1997; Ciliberto et al., 2017) and Multi-domain Learning (MDL) (Joshi et al., 2012). In fact, domains can be seen as tasks in MTL or MDL. However, an explicit domain label is provided and annotated examples are available for each task. A particularly challenging and useful setting in practice is Unsupervised Domain Adaptation (UDA) (Tzeng et al., 2017; Ganin et al., 2016), in which labeled samples from a source domain are used together with unlabeled samples from the target domain to improve performance on the latter.

This work focuses on Source-Free Unsupervised Domain Adaptation (SF-UDA) (Liang et al., 2020) for the image classification task. SF-UDA is a two-steps sequential version of UDA in which the source-domain labeled data is only accessible in the first training phase. Adaptation to the new domain is carried out in a second stage where only the unlabeled data from the target domain is available. SF-UDA nicely matches applications where continual adaptation is required with computational and memory constraints, or where privacy policies prevent access to the source data. Since

these techniques are usually applied to fine-tune and adapt models with pre-trained weights, there are two different transfers into play: (1) from the base task (used for pre-training) to the source domain, and (2) from the source domain to the target domain: we refer to this combined transfer as *double-transfer*. The objective of this work is two-fold. Firstly, we aim to study the impact of the main design choices in SF-UDA approaches, such as the backbone architecture, pre-training dataset, and the way double-transfer is performed. Secondly, we aim to investigate the strengths and failure modes of SF-UDA methods, also comparing them with standard UDA approaches.

Recent works (Liang et al., 2020; Ding et al., 2022) show that SF-UDA techniques achieve comparable performance with state-of-the-art UDA methods on common benchmarks. In contrast, Kim et al. (2022) reports that recent UDA methods perform well on standard benchmarks because they *overfit* the task. Indeed, when employed in other settings (i.e., non-standard architectures or datasets) they result in worse accuracy than previous methods. We investigate with targeted experiments whether overfitting affects recent SF-UDA methods as well. We pursue such objectives through large-scale systematic experiments encompassing more than 500 different architectures on 6 separate domain adaptation datasets, totaling 23 domains and 74 domain shifts. We employ different probing and SF-UDA methods to better analyze the functioning of double-transfer methods, providing a ready-to-use recipe for effective system design. Our main findings are as follows:

- The pre-training dataset choice and the resulting accuracy on the ImageNet top-1 benchmark directly impacts on the domain generalization and SF-UDA performance, for both CNNs and Vision Transformers.

- Most SF-UDA methods fine-tune the model on the source domain before adaptation. However, we show that in some cases this causes severe performance degradation. Specifically, we identify the type of normalization layers as having a critical role in this context.

- Besides fine-tuning, the normalization strategy heavily affects the failure rate of SF-UDA in general. We present a large-scale analysis of SF-UDA methods' failure rates, comparing architectures with Layer Normalization (LN) (Ba et al., 2016) and Batch Normalization (BN) (Ioffe & Szegedy, 2015) and attesting the improved robustness of the former.

- SF-UDA methods, like SHOT (Liang et al., 2020), SCA (see Sec. 3.2), and NRC (Yang et al., 2021a), perform well also with architectures and datasets different from the usual benchmark ones and are competitive with state-of-the-art UDA methods.

The full list of architectures, results in csv format, the code and pre-trained weights will be released[1].

## 2 BACKGROUND AND RELATED WORK

Let $\mathcal{X}$ be the input space, e.g., the image space, $\mathcal{Z} \subseteq \mathbb{R}^D$ a representation space, i.e., the feature space, and $\mathcal{Y} = \{1, \ldots, C\}$ the output space for multi-class classification. A *feature extractor* (backbone) is a function $f_{\boldsymbol{\theta}} : \mathcal{X} \to \mathcal{Z}$ with parameter $\boldsymbol{\theta}$, while a *classifier* is a function with parameter $\boldsymbol{\phi}$ that assigns a label to any feature vector, $h_{\boldsymbol{\phi}} : \mathcal{Z} \to \mathcal{Y}$. We introduce two data distributions over $\mathcal{X} \times \mathcal{Y}$: $\mu_S$ that models the source domain and $\mu_T$ that models the target domain.

**Domain Generalization (DG).** In this work, we consider DG setting as the reference task to investigate transferability among domains. First introduced by Blanchard et al. (2011), its goal is to find a feature extractor $f_{\boldsymbol{\theta}}$ and a classifier $h_{\boldsymbol{\phi}}$ from $N$ *i.i.d.* samples of a given domain (the source $\mu_S$) that will perform well on other unseen domains (in our case just the target domain $\mu_T$), that is:

$$\min_{\boldsymbol{\theta}, \boldsymbol{\phi}} \mathbb{E}_{(\boldsymbol{x}, y) \sim \mu_T} [\mathbb{1}\{h_{\boldsymbol{\phi}}(f_{\boldsymbol{\theta}}(\boldsymbol{x})) \neq y\}] \qquad \text{given } (\boldsymbol{x}_i, y_i)_{i=1}^N \sim \mu_S^N, \tag{1}$$

where $\mathbb{1}\{\text{condition}\}$ is the indicator function. In this setting, some assumptions on the relationship between tasks ($\mu_S$ and $\mu_T$) are needed, but only data sampled from the source domain can be used for training, while target domain data is accessible at test time only. We remark that this is similar to the standard supervised-learning problem of generalization (Gen), with the exception that, here, the training and test distributions are different. Several methods specifically target this problem, e.g., (Volpi et al., 2018; Arjovsky et al., 2019; Ilse et al., 2020); see Wang et al. (2021) for a review.

---

[1] https://www.github.com/anonymous_author/double_transfer/

**Usupervised Domain Adaptation (UDA).** As for DG, the final goal of UDA is to learn a model that performs well on the target domain. However, differently from DG, more information is available: unlabeled samples from the target (marginal) distribution are accessible together with the labeled data from the source. Since the seminal theoretical works (Ben-David et al., 2006; 2010; Mansour et al., 2009), many UDA methods have been proposed in the literature for image classification, including adversarial training (Ganin et al., 2016), bidirectional matching (Na et al., 2021), per-class kernel mean discrepancy minimization (Kang et al., 2019), and also for other tasks such as object detection (Oza et al., 2021) and semantic segmentation (Toldo et al., 2020).

**Source-Free Unsupervised Domain Adaptation (SF-UDA).** SF-UDA is similar to UDA, but it is more constrained. The learning process is divided into 2 phases: the (labeled) source data is available only in the first training step. Then, adaptation to the new domain occurs in the second stage, where only the unlabeled target data is available. The problem can be formulated as:

$$\min_{\boldsymbol{\theta}, \boldsymbol{\phi}} \mathbb{E}_{(\boldsymbol{x}, y) \sim \mu_T} [\mathbb{1}\{h_{\boldsymbol{\phi}}(f_{\boldsymbol{\theta}}(\boldsymbol{x})) \neq y\}] \tag{2}$$

$$\text{given } (\boldsymbol{x}_i, y_i)_{i=1}^N \sim \mu_S^N \text{ (only at step 1)} \qquad \text{and } (\boldsymbol{x}_j)_{j=1}^M \sim \mu_{T(\mathcal{X})}^M \text{ (only at step 2)}, \tag{3}$$

where $\mu_{T(\mathcal{X})}$ is the marginal of $\mu_T$ over the input space $\mathcal{X}$ and $M$ is the number of available target samples. We remark that, even if it is not possible to share data among the 2 steps, model parameters $\boldsymbol{\theta}'$ and $\boldsymbol{\phi}'$ found at step 1 are accessible at step 2. Recently, SF-UDA has gained significant interest: since the work by Liang et al. (2020), many other methods have been proposed and they achieved remarkable results on standard UDA benchmarks (Li et al., 2020; Yang et al., 2021b; Kundu et al., 2020; Huang et al., 2021; Kurmi et al., 2021; Xia et al., 2021; Chu et al., 2022; Ding et al., 2022; Kundu et al., 2022). In Sec. 3.2, we will illustrate some SF-UDA methods in greater detail.

**Experimental Studies.** Training DNNs is demanding in terms of computational resources, time and data. Hence, there is great interest in the scientific community into understanding how to obtain representations that can be conveniently transferred to new tasks and what are the key ingredients to build more efficient architectures and training methods. For this reason, several studies on transfer learning (i.e., fine-tuning for classification or different vision tasks) have been conducted, such as (Chu et al., 2016), (Huh et al., 2016), and (Kornblith et al., 2019). Concerning UDA, (Zhang & Davison, 2020) studies model selection for classical methods (and for CNNs) based on the accuracy achieved on ImageNet, while (Kim et al., 2022) provides an analysis of different pre-training techniques for DG and UDA. In this work, instead, we target SF-UDA approaches. SF-UDA is highly relevant for applications, since it allows the design of efficient algorithms while, as we show in our experiments, achieving comparable performance to UDA. In our analysis, we decouple the effects on the final result of the two adaptations into play in double-transfer: from pre-training to source domain and from source to target domain. With our results, we provide best practices for robust SF-UDA pipelines. Our empirical analysis includes more than 500 architectures (including both CNNs and Vision Transformers). We test them on 6 datasets for domain adaptation (for a total of 74 domain shifts). To the best of our knowledge, this is the most extensive study on SF-UDA.

## 3 METHODS

### 3.1 PROBING

For our initial experiments we adopt two probing methods to evaluate the quality of features extracted from the models: linear probing and cluster probing.

**Linear Probing (LP).** The pre-trained feature extractor is fixed and employed to compute features from training images. These are employed to train a linear classifier (i.e., a multinomial regressor). Finally, classification accuracy is evaluated on the test set. This method is commonly used in the literature to evaluate feature extractors pre-trained with self-supervision (Chen et al., 2021).

**Cluster Probing (CP).** As in LP, we fix the feature extractor and compute features from the training images. Then, for each class, the *class prototype* is computed as the average feature vector of the examples from that class. At prediction time, new samples are classified based on the class of the closest prototype (in our case, we use the cosine dissimilarity). As in LP, we evaluate generalization capabilities on the test set. CP allows inspecting the properties of the learned representation, providing useful insights for SF-UDA which exploits the underlying structure of the feature space to overcome the lack of ground truth on the target domain. Refer to App. A for more details.

**Remark.** In next sections, we append "Gen" or "DGen" to the method name to indicate the test accuracy on the source or target domain, respectively. If nothing is indicated we use LP as default.

## 3.2 SF-UDA

**Simple Class Alignment (SCA).** This algorithm adapts the classifier to the target domain without altering the feature extractor. As in *cluster probing*, a first phase computes class *prototypes* of labeled data of the source domain. In the second step, the prototypes are used as initialization centroids in spherical k-means (Hornik et al., 2012), which is executed on the target unlabeled data. Hence, the final centroids are adapted prototypes that account for the domain shift. The resulting classifier assigns the class of the closest centroid to new inputs (based on cosine dissimilarity). This method was presented in (Kang et al., 2019) to compute pseudo-labels. Variants are discussed in App. B.

**Source HypOthesis Transfer (SHOT).** This algorithm (Liang et al., 2020; 2021) is considered the state-of-the-art in SF-UDA for its efficiency and is a solid baseline for novel SF-UDA methods. The first transfer requires fine-tuning the model to the source domain. The second transfer alternates two steps iteratively: (1) pseudo-labels computation for target samples, and (2) feature extractor fine-tuning using the Information Maximization loss on previously computed pseudo-labels, while keeping the classifier fixed. Interestingly, the latter proved crucial for achieving high performance. For pseudo-labels computation, SHOT builds on a modified version of SCA. The feature extractor ($f_\theta$) and the classifier ($h_\phi$), trained on the source domain, are used to predict probabilities for each target sample $\boldsymbol{x}_i$ for all the $C$ classes $\boldsymbol{p}_i = (p_{i1}, \ldots, p_{iC})$. Then, the initialization prototype for class $c, c = 1, \ldots, C$ is computed as a weighted average of the target features:

$$\boldsymbol{k}_c = \frac{\sum_{\boldsymbol{x}_i \in S_{tgt}} p_{ic} f_\theta(\boldsymbol{x}_i)}{\sum_{\boldsymbol{x}_i \in S_{tgt}} p_{ic}}. \tag{4}$$

The prototypes $\boldsymbol{k}_1, \boldsymbol{k}_2, \ldots, \boldsymbol{k}_C$ are used to initialize spherical k-means (as in SCA), and the final 1-NN classifier computes the pseudo-labels of the target dataset.

**Remark.** We select SCA and SHOT for our experiments since they are recent representative SF-UDA approaches that tackle the second transfer in two complementary ways. Indeed, while SHOT keeps the classifier fixed and adapts the feature extractor, SCA does exactly the opposite. Further experiments with the NRC method (Yang et al., 2021a) corroborate our results.

## 4 EXPERIMENTS

### 4.1 SETUP

**Models.** We rely on the *PyTorch Image Models* Python library (timm) (Wightman, 2019) for the implementation of the various models considered in our experiments. It provides access to a remarkably large number of different architectures and pre-trained weights. In our evaluation of the probing approaches and SCA (without fine-tuning) we used 500 models taken from more than 25 different families of architectures (e.g., VGG (Simonyan & Zisserman, 2015), ResNet (He et al., 2016), EfficientNet (Tan & Le, 2019), ConvNext (Liu et al., 2022), ViT (Dosovitskiy et al., 2020), SWIN (Liu et al., 2021), Deit (Touvron et al., 2021) and XCiT (Ali et al., 2021)). Instead, for the experiments with SHOT and with networks fine-tuning, we sampled a subset of 59 models, taken from more than 12 families of architectures. Notably, our analysis comprises both modern Vision Transformers and more traditional CNNs. More details are available in App. C.

**Pre-training.** For the first transfer, we consider two datasets: ImageNet (ILSVRC-2012), composed of 1.2M images for 1000 mutually exclusive classes, and the superset ImageNet21k (Deng et al., 2009), composed of 14M images for 21,841 not mutually exclusive classes. Specifically, we either consider models pre-trained on ImageNet (*IN*) or on ImageNet21k and then fine-tuned on the 1000 classes of ImageNet (*IN21k*).

**Domain Adaptation Image Datasets.** DA datasets for image classification typically have two or more sub-datasets corresponding to different domains sharing the same classes. In our experiments, we considered: *DomainNet* (Peng et al., 2019), a large dataset with 6 domains of common objects, divided into 345 categories (596 006 images); *ImageClef-DA* (Long et al., 2017), a small dataset with 4 domains and 12 classes (2 400 images); *Modern Office-31* (Ringwald & Stiefelhagen, 2021), a novel version of the Office-31 dataset (Saenko et al., 2010) with an additional synthetic domain (for a total of 4) and 31 classes (7 210 images); *Visda-2017* (Peng et al., 2017), a large sim-to-real dataset with 12 categories and 2 domains (280 000 images); *Office-Home* (Venkateswara et al., 2017), with 4 domains and 65 categories (15 588 images); *Adaptiope* (Ringwald & Stiefelhagen, 2021), having 3 domains (*synthetic*, *product* and *life*) and 123 different classes (36 900 images). We end up with

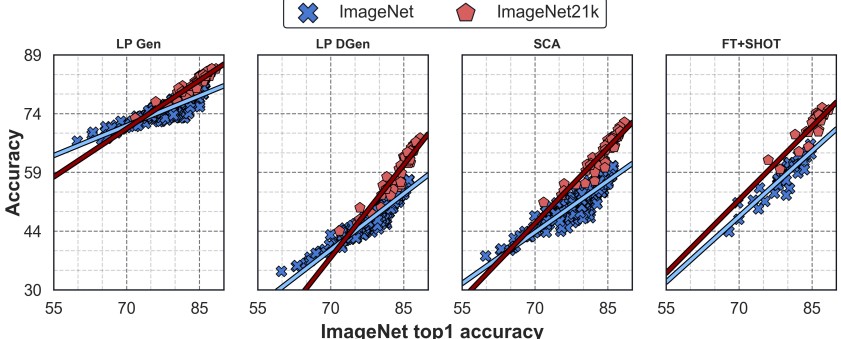

**Figure 1:** We compare performance of different feature extractors (which have been pre-trained either on ImageNet or Imagenet21k) for, respectively: (i) Generalization LP, (ii) Domain generalization LP, (iii) SCA and (iv) FT + SHOT. In the first sub-plot (LP Gen) the test accuracy on the source is considered, while on other subplots (LP DGen, SCA and FT+SHOT) the test is performed on the target domain.

**Table 1:** Adjusted $\bar{R}^2$ values for linear (only top1 accuracy considered) and multi-linear (top1 and pre-training) explanatory models for linear probing (LP), cluster probing (CP), SHOT, SCA and fine tuning (FT).

|  | LP Gen | CP Gen | LP DGen | CP DGen | SCA | SHOT | FT Gen | FT DGen | FT+ SCA | FT+ SHOT |
|---|---|---|---|---|---|---|---|---|---|---|
| **Linear** | 0.736 | 0.756 | 0.810 | 0.755 | 0.731 | 0.792 | 0.803 | 0.698 | 0.668 | 0.838 |
| **Multi-Linear** | 0.851 | 0.908 | 0.935 | 0.918 | 0.902 | 0.890 | 0.878 | 0.822 | 0.792 | 0.932 |

23 different domains and 74 domain pairs in total[2].

**Experimental Details.** We use the aforementioned datasets to conduct a systematic study on the main design choices in *double-transfer*. Specifically, to evaluate the generalization capability (Gen) of the various feature extractors (first transfer), we randomly split the images from each domain into a training (80%) and a test set (20%). We use the former to either train the classifier (LP or CP) or to fine-tune the model and we evaluate the accuracy on the test set. The final performance of one feature extractor is given by the average of all the accuracy values obtained on the 23 domains. Similarly, for the second transfer (DGen and SF-UDA settings), we consider each domain pair and the final performance is obtained by averaging over all the 74 pairs. Note that for SF-UDA we consider different combinations for the two transfers. For the first one, we either train a classifier on the source domain, keeping the feature extractor fixed (as in LP and CP), or we fine-tune (FT) it while training the classifier. Instead, for the second phase we mainly consider SHOT and SCA. This leads to 4 different combinations: SCA, FT+SCA, SHOT and FT+SHOT. Finally, in SF-UDA, as it is common in the literature and in UDA benchmarks, we consider the transductive setting (Kouw & Loog, 2019) where the accuracy is evaluated on the same images used for adaptation (albeit without labels). For technical details on the experiments refer to App. D.

## 4.2 RELEVANCE OF PRE-TRAINING AND IMAGENET ACCURACY IN SF-UDA

*How to choose the best feature extractor for SF-UDA?*

In Fig. 1, we compare the LP performance of different feature extractors for Generalization and Domain Generalization. Furthermore, we present results for SCA and FT+SHOT. Both ImageNet and ImageNet21k pre-trainings are considered. We plot the relationship between the accuracy obtained on ImageNet and the one achieved on the task of interest. As it can be noticed, the latter is mainly influenced by two factors: (i) the ImageNet top-1 accuracy, and (ii) the pre-training dataset. Specifically, in all cases the performance on the task of interest depends linearly on the ImageNet top-1 accuracy of the backbone. For instance, the accuracy of a backbone $\mathcal{B}$ can be modeled as:

$$accuracy(\mathcal{B}) = m \cdot top1(\mathcal{B}) + q + \epsilon, \tag{5}$$

---

[2]In Visda-2017, differently from the common benchmark, we consider both experiments (synthetic $\rightarrow$ real and real $\rightarrow$ synthetic).

**Table 2:** ImageNet vs ImageNet21k pre-training. The accuracy reported are the average of 74 domain shifts.

| | Backbone | VGG19 | ResNet50 | W-ResNet50 | DenseNet161 | ConvNext B |
|---|---|---|---|---|---|---|
| | # params (MM) | 143.7 | 25.6 | 68.9 | 28.7 | 88.6 |
| LP DGen | IN | 45.2 | 47.2 | 50.5 | 48.0 | 54.4 |
| | IN21k | 47.6 | 51.3 | 52.5 | 52.2 | 65.2 |
| SCA | IN | 49.4 | 49.8 | 53.3 | 52.7 | 58.4 |
| | IN21k | 53.1 | 58.2 | 58.2 | 58.2 | 68.5 |
| FT+SHOT | IN | 56.0 | 55.9 | 62.1 | 61.6 | 65.1 |
| | IN21k | 55.7 | 62.0 | 64.0 | 65.3 | 72.7 |

where $m$ and $q$ are the parameters of the linear model (specific to each experiment), while $\epsilon$ is a random variable that accounts for the variance in the data, not explained by the model.

Instead, regarding the dataset choice for pre-training, it is known from the literature (Dosovitskiy et al., 2020) that using ImageNet21k can boost the accuracy on ImageNet. Thus, according to Equation 5 this should also improve the accuracy for all the downstream tasks. This is confirmed by our experiments. Notably, by taking two models with the same ImageNet top-1 accuracy but pre-trained on different datasets, it is evident from Fig. 1 that the model pre-trained on ImageNet21k has better performance (on average). This implies that the ImageNet21k pre-training yields an additional improvement in the considered transfer tasks that cannot be explained by the increased ImageNet accuracy of the model. To account for this, we introduce the pre-training into the linear statistical model and describe the data through a multi-linear model with interaction. Let $pretrain$ be equal to 1 for ImageNet21k backbones and to 0 for ImageNet backbones, then the model becomes:

$$accuracy(\mathcal{B}) = [m + \Delta m \cdot pretrain(\mathcal{B})] \cdot top1(\mathcal{B}) + q + \Delta q \cdot pretrain(\mathcal{B}) + \epsilon, \qquad (6)$$

where $\Delta m$ and $\Delta q$ are the newly introduced parameters in the model.

In Tab. 1, we compare the goodness-of-fit (adjusted $\bar{R}^2$) of the multi-linear model with the linear one. The difference is significant for all experiments with probing and SF-UDA methods. App. E reports the coefficients of the statistical models and the scatter plots of further experiments. In addition, in App. F we compare the performance of the same architecture (ResNet50) pre-trained on ImageNet21k in 2 ways: (i) with the standard training process (used for all experiments in this work), and (ii) with the technique proposed by Ridnik et al. (2021), which employs a specific semantic pre-training on a filtered version of ImageNet21k to reduce the strong class imbalance and to tackle the multi-label nature of the dataset. We observe that semantic pre-training significantly improves domain generalization, while when applying SF-UDA methods the gap is recovered and the model performs equally well with both pre-trainings. In App. J, we compare supervised and self-supervised pre-training strategies. Despite the promising results, the former still outperform the second.

***Does the size of the model count?*** One may expect that only large models (in terms of number of parameters) could benefit from such a large pre-training dataset as ImageNet21k. However, in Tab. 2 we show that this also applies to smaller models.

## 4.3 FINE-TUNING ON THE SOURCE DOMAIN

In this section, we study the impact on downstream tasks of fine-tuning the model on the source data (first transfer). Results for 59 models are reported in Fig. 2 averaged over 23 domains in case of experiments with Gen tasks and on the 74 domain pairs in all other cases. The reported plots present accuracy differences between each method and its baseline (as green or red arrows). For instance, the top-left plot reports the difference in source accuracy between no fine-tuning (i.e., LP) and fine-tuning on the source domain.

Firstly, it can be noticed that in cases where no fine-tuning is applied (namely SHOT and SCA plots), a remarkable accuracy gain is still obtained with respect to LP DGen. Interestingly, Tab. 3 highlights how SHOT (which adapts the feature extractor on the target domain) obtains an accuracy gain of $6.97\%(\pm 2.24)$ on average on the target domain. In comparison, SCA gains on average $4.21\%(\pm 1.39)$, while being 50-60 times faster than SHOT (see App. H).

Then, we consider the case where fine-tuning is applied on the source domain. Contrary to the Gen task case where fine-tuning the feature extractor on one domain increases its However, this does not always improve performance on the second transfer, as can be noticed in the second column of

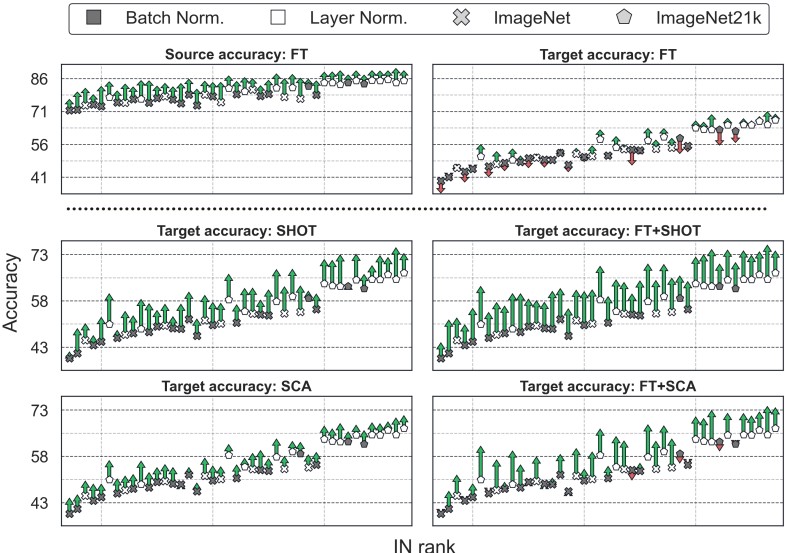

**Figure 2:** Architectures are sorted on the x-axis according to IN top1 accuracy (best ones on the right). We distinguish models based on the employed pre-training and normalization layers. The markers' y-value represents the LP accuracy of the model, while the green/red arrows represent the accuracy improvement/degradation when a method is applied. FT and LP are always performed on the source with labels, while adaptation methods are always performed on the target without labels. **Top-left**: source accuracy improves with fine-tuning (average over 23 domains). **Top-right**: target accuracy slightly increases for layer-norm models but it is degraded for batch-norm models. **Bottom**: SCA and SHOT improve after fine-tuning. In both cases we average results over 74 domain pairs (we report a subset of the models for visualization purposes).

Fig. 2. Specifically, while for SHOT fine-tuning during the first transfer always yields a subsequent performance gain, models with BN layers do not benefit from this adaptation on the source for SCA and especially for the FT-only target accuracy. In some cases, fine-tuning on the source even degrades performance (see App. I for further results).

We thoroughly analyze this matter in Fig. 3. We report results for domain pairs from ModernOffice31 (namely, Synthetic and DSLR), for which this phenomenon is most evident. However, it occurs also in a large number of other domain pairs. In the first row, we experiment in the Synthetic → DSLR case. As evidenced by the first-column plots, BN layers in this case lead to a very significant target accuracy degradation after fine-tuning compared to naïve LP. Nonetheless, in the second column we show that using ADABN (Li et al., 2017) to adapt the BN layers on the target domain compensates for the initial loss. Concerning SF-UDA, note that while SCA is strongly affected by the performance degradation brought by BN layers adaptation (third column), SHOT is able to nicely recover (fourth column), since it employs BN statistics computed on the target domain during the iterative feature extractor adaptation. Finally, in the second row we report results for the DSLR → Synthetic transfer. Notably, in this case ADABN does not help recover performance, while both SCA and SHOT allow to compensate for the initial loss. Note that there may exist methods to improve the target accuracy after fine-tuning of models with BN (e.g., fixing BN statistics, using augmentations, etc.), but the study of these techniques is out of the scope of our analysis.

**Remark 1.** One could ascribe the performance drop to architectural properties other than the BN layers, but with the previous examples (i.e., column 2 in Fig. 3) we show that the main cause actually lies in the BN statistics. Moreover, the issue never occurs for models with LN layers.

**Remark 2.** The performance drop when fine-tuning models with BN layers is especially evident for pairs with shifts between real and synthetic domains, but it is not limited to such cases (see, e.g., Clipart → Art (OfficeHome) and Webcam → Amazon (ModernOffice31)). In App. G we show a more extensive comparison between models with BN and LN layers.

**Remark 3.** Tab. 3 reports the average difference in accuracy with respect to LP DGen, grouping the models used for this experiment by pre-training dataset and normalization layers. We compare performance for FT DGen, SCA, FT + SCA, SHOT, and FT + SHOT.

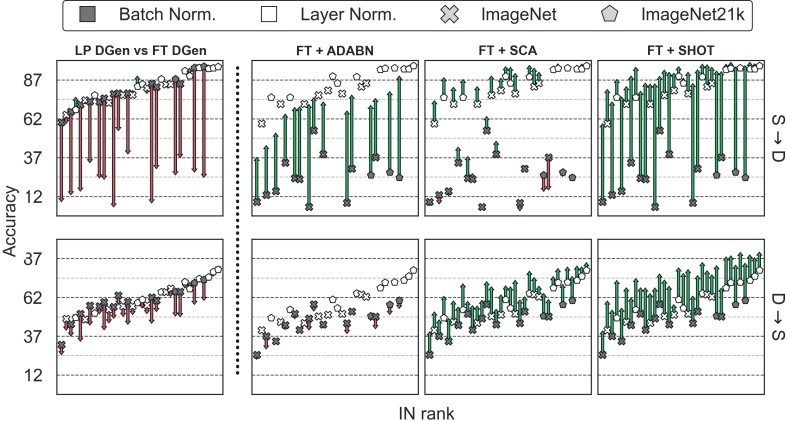

**Figure 3:** Rows represent different domain pairs (Sythetic → DSLR and DSLR → Synthetic from Modern Office31). In column 1, arrows indicate the effect of FT on the target accuracy with respect to the LP DGen baseline (the markers). In the other columns, the markers represent the target domain accuracy after fine-tuning and arrows represent the performance difference using ADABN, SCA or SHOT (after the fine-tuning).

**Table 3:** Average accuracy change (%) with respect to DGen with LP. For each architecture, the accuracy change is the average over 74 different domain-shifts. Each row, shows the performance obtained by models grouped by different combinations of their pre-training and the normalization layers. The number of considered models is reported in parenthesis. In each column, we report results for different downstream tasks.

| Models | FT DGen | SCA | SHOT | FT+SCA | FT+SHOT |
|---|---|---|---|---|---|
| **All** (59) | 0.46 ± 3.71 | 4.21 ± 1.39 | 6.97 ± 2.24 | 4.90 ± 3.86 | 9.66 ± 1.83 |
| **IN** (32) | -0.39 ± 3.23 | 3.97 ± 1.40 | 6.57 ± 1.95 | 4.20 ± 3.62 | 9.68 ± 1.76 |
| **IN21k** (27) | 1.46 ± 4.04 | 4.49 ± 1.35 | 7.45 ± 2.51 | 5.72 ± 4.03 | 9.63 ± 1.93 |
| **BN** (32) | -2.21 ± 2.75 | 4.08 ± 1.55 | 6.21 ± 2.39 | 2.16 ± 2.94 | 9.68 ± 1.96 |
| **LN** (27) | 3.62 ± 1.57 | 4.36 ± 1.19 | 7.88 ± 1.68 | 8.14 ± 1.66 | 9.63 ± 1.69 |
| **IN+BN** (24) | -1.61 ± 2.66 | 3.75 ± 1.52 | 6.29 ± 2.10 | 2.74 ± 2.91 | 9.67 ± 1.90 |
| **IN+LN** (8) | 3.28 ± 1.55 | 4.64 ± 0.66 | 7.42 ± 1.12 | 8.59 ± 0.98 | 9.71 ± 1.40 |
| **IN21k+BN** (8) | -4.02 ± 2.30 | 5.07 ± 1.27 | 5.97 ± 3.29 | 0.44 ± 2.45 | 9.72 ± 2.28 |
| **IN21k+LN** (19) | 3.77 ± 1.59 | 4.25 ± 1.35 | 8.07 ± 1.86 | 7.95 ± 1.86 | 9.60 ± 1.84 |

## 4.4 ROBUSTNESS AND FAILURES ANALYSIS

In SF-UDA, target annotations are unavailable, precluding performance evaluation after adaptation in real scenarios. Therefore, it would be beneficial in practice to have an estimate of how likely SF-UDA methods are to degrade performance on the target domain and how severe such degradation could be. To this aim, we extensively evaluate the robustness of SF-UDA. Given a model and a DA task, we fix LP DGen as baseline and consider an SF-UDA approach to fail if its target accuracy after adaptation is lower than the baseline's. Tab. 4 groups 59 models by pre-training dataset and normalization layers and reports the average failure rates of each group on the 74 considered domain pairs. BN models always show a remarkably higher failure rate: FT degrades the target accuracy 51.5% of the times for BN models, while just 28.1% for LN ones. Applying SCA or SHOT after fine-tuning can reduce failures, but LN models are still more robust and fail less frequently. Pre-training also impacts the failure rate: LN models pre-trained on ImageNet21k have a lower rate than those pre-trained on ImageNet. However, BN models deteriorate more often when pre-trained on ImageNet21k. Indeed, ImageNet21k models, as shown in Sec. 4.2, achieve a better LP DGen, and therefore a stronger baseline value. For this reason, the instability of BN layers might result in a more apparent degradation. Finally, note that SCA is very sensitive to the normalization layers choice and that FT+SHOT is the method with the lowest failure rate overall.

**Performance Degradation in Case of Failure.** As before, degradation magnitude mostly depends on normalization: in case of failure, performance on target domain after FT (on source) decreases by 10.5% for BN and by just 2.10% for LN. SCA is unable to recover the FT degradation: in fact, the accuracy of FT+SCA decreases by 12.7% for BN and only 1.4% for LN models. The full table with average performance degradation is reported in App. G.

**Table 4:** Average Failure Rate (%) on 74 domain shifts. Each row, shows the performance obtained by models grouped by different combinations of their pre-training and the normalization layers. The number of considered models is reported in parenthesis. In each column, we report results for different downstream tasks.

| Models | FT DGen | SCA | SHOT | FT+SCA | FT+SHOT |
|---|---|---|---|---|---|
| **All** (59) | $40.82 \pm 16.14$ | $21.05 \pm 11.92$ | $11.18 \pm 8.75$ | $19.08 \pm 14.57$ | $4.47 \pm 3.61$ |
| **IN** (32) | $45.86 \pm 13.19$ | $25.84 \pm 12.43$ | $13.13 \pm 8.97$ | $23.35 \pm 13.89$ | $5.15 \pm 3.62$ |
| **IN21K** (27) | $34.83 \pm 17.48$ | $15.37 \pm 8.42$ | $8.86 \pm 8.03$ | $14.01 \pm 13.93$ | $3.65 \pm 3.49$ |
| **BN** (32) | $51.52 \pm 11.42$ | $27.24 \pm 11.64$ | $14.82 \pm 9.32$ | $29.77 \pm 11.05$ | $5.83 \pm 3.77$ |
| **LN** (27) | $28.13 \pm 10.85$ | $13.71 \pm 7.25$ | $6.86 \pm 5.61$ | $6.41 \pm 4.52$ | $2.85 \pm 2.69$ |
| **IN+BN** (24) | $50.51 \pm 11.26$ | $28.89 \pm 12.78$ | $14.13 \pm 9.11$ | $29.00 \pm 11.19$ | $5.52 \pm 3.74$ |
| **IN+LN** (8) | $31.93 \pm 7.61$ | $16.72 \pm 4.62$ | $10.14 \pm 8.36$ | $6.42 \pm 2.58$ | $4.05 \pm 3.23$ |
| **IN21k+BN** (8) | $54.56 \pm 12.11$ | $22.30 \pm 5.11$ | $16.89 \pm 10.27$ | $32.09 \pm 11.00$ | $6.76 \pm 3.96$ |
| **IN21k+LN** (19) | $26.53 \pm 11.77$ | $12.45 \pm 7.86$ | $5.48 \pm 3.39$ | $6.40 \pm 5.19$ | $2.35 \pm 2.33$ |

**Table 5:** Comparison, using modern architectures (SWIN and ConvNext) of SF-UDA (SCA, SHOT and NRC) with not Source-Free methods: DANN (Ganin et al., 2016), CDAN (Long et al., 2016), AFN (Xu et al., 2019), MDD (Zhang et al., 2019) and MCC (Jin et al., 2020).

| Backbone | Dataset | DANN[†] | CDAN[†] | AFN[†] | MDD[†] | MCC[†] | FT+SCA[*] | FT+SHOT[*] | FT+NRC[*] |
|---|---|---|---|---|---|---|---|---|---|
| SWIN L | Office-Home | 86.6 | 88.4 | 85.7 | 86.5 | 88.3 | 86.4 | 89.3 | 89.5 |
| | DomainNet | 49.4 | 50.4 | 46.4 | 41.5 | 47.1 | 49.0 | 51.2 | 48.4[◇] |
| ConvNext XL | Office-Home | 86.7 | 89.5 | 85.5 | 86.4 | 88.9 | 87.4 | 88.8 | 89.2 |
| | DomainNet | 48.8 | 51.2 | 46.7 | 42.8 | 45.6 | 49.8 | 51.0 | 47.1[◇] |

[†] Requires source, reproduced by Kim et al. (2022).
[*] Source-Free, reproduced by us.
[◇] Trained and evaluated on a random 15% subset of DomainNet (official distributed implementation not available).

## 4.5 Comparison with Standard UDA

Recent results indicate overfitting to the benchmark for UDA approaches (Kim et al., 2022): with modern backbones, classic algorithms outperform newer ones on uncommon datasets. In constrast, SF-UDA techniques demonstrate a performance gain with respect to the LP DGen baseline.
In Tab. 5, we report SCA, SHOT, and NRC in a setting different from the standard evaluation protocol and compare them to both established and novel UDA solutions. We consider two modern architectures: SWIN (Liu et al., 2021) and ConvNext (Liu et al., 2022). Three main observations arise: (i) despite the stricter constraints on the learning setting, SF-UDA algorithms are competitive with standard UDA; (ii) SCA performs on par (or even outperforms) common UDA algorithms while being more efficient (see Tab. 12 in App. H) and, (iii) although FT+NRC has been trained on a subset of DomainNet, this proved sufficient for it to be competitive with UDA counterparts.

## 5 Conclusions

The lack of supervision in UDA hinders model selection when designing and deploying learning systems. In this challenging setting, large-scale experiments can provide empirical insights into best practices to tackle real-world UDA problems. In this work, we shed light on the role of the key design choices underlying SF-UDA algorithms: pre-training dataset, architecture, fine-tuning, and normalization strategy. To this end, we present extensive experiments evaluating the impact of such choices. Our results consistently demonstrate that pre-training accuracy on the standard ImageNet strongly correlates with domain adaptation performance. Moreover, despite similar performance on the standard ImageNet test set, models pre-trained on ImageNet21k adapt more effectively. Our results challenge the common practice of fine-tuning to the task at hand: blind adaptation might lead to catastrophic failures, mainly when BN is applied and normalization statistics of the source data are distant from the target ones. Our robustness analysis confirms the critical role of the normalization layers. On average, BN architectures suffer from a higher failure rate, i.e., adaptation deteriorates performances. Despite being more constrained in terms of source data availability and computational cost, SF-UDA proves competitive with UDA approaches. Our analysis of uncommon architectures and domain shifts demonstrates the robustness of SF-UDA beyond standard benchmarks.

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

# A PROBING METHODS

In this section, we discuss the probing methods employed to inspect and assess learnt models.

We follow the the same notation introduced in Sec. 2. Let $f_{\boldsymbol{\theta}}$ be a given fixed feature extractor and $C$ the number of classes. We introduce the training set $\mathcal{S}_{\text{TRAIN}} = \{(\boldsymbol{z}, y)\}_{i=1}^{N}$, where $\boldsymbol{z}_i = f_{\boldsymbol{\theta}}(\boldsymbol{x}_i)$ and $(\boldsymbol{x}_i, y_i) \sim \mu_S$ are *i.i.d.* samples from the source domain, for $i = 1, \ldots, N$.

Similarly, we define $\mathcal{S}_{\text{TEST}}$ to be a set of $M$ *i.i.d.* evaluation samples. Based on the experiment, evaluation samples can come from the same distribution of the training (generalization) or from a different one (domain-generalization and domain adaptation).

**Linear Probing** (LP) trains and evaluates a linear classifier (Multinomial Regression) on the features extracted from a fixed model. During training of the linear classifier, we seek for a function $g$ that maps the features to the $C - 1$ simplex minimizing the (L2-regularized) log loss:

$$g : \mathcal{Z} \to \Delta^{C-1},$$
$$\boldsymbol{z} \mapsto \sigma(\boldsymbol{W}\boldsymbol{z}),$$

where $\sigma$ is the *softmax* function and $\boldsymbol{W} \in \mathbb{R}^{C \times D}$ is found as:

$$\boldsymbol{W} = \underset{\boldsymbol{W}' \in \mathbb{R}^{C \times D}}{\arg \min} \lambda ||\boldsymbol{W}'||_F^2 + \frac{1}{|\mathcal{S}_{\text{TRAIN}}|} \sum_{(\boldsymbol{z}, y) \in \mathcal{S}_{\text{TRAIN}}} - \log(\sigma(\boldsymbol{W}'\boldsymbol{z})_y),$$

with $\lambda > 0$ regularization hyperparameter.
Then, we can convert this function to a classifier $h_{MR}$:

$$h_{MR} : \mathcal{Z} \to \mathcal{Y},$$
$$\boldsymbol{z} \mapsto \underset{k}{\arg \max}\, g(\boldsymbol{z})_k.$$

Hence, linear probing accuracy is evaluated on the test set:

$$\text{LP accuracy} = \frac{1}{|\mathcal{S}_{\text{TEST}}|} \sum_{(\boldsymbol{z}, y) \in \mathcal{S}_{\text{TEST}}} \mathbb{1}\{h_{MR}(\boldsymbol{z}) = y\}. \tag{7}$$

**Cluster Probing** (CP): trains and evaluates a 1 Nearest Neighbour (1-NN) classifier on the features extracted from a fixed model. For every class, a prototype is found averaging the feature vectors of that class and, then, the prototypes are used as a 1-NN classifier. In particular, for each class $c$ the prototype is evaluated as:

$$\boldsymbol{k}_c = \frac{\sum_{(\boldsymbol{z}, y) \in \mathcal{S}_{\text{train}}} \mathbb{1}\{y = c\}\boldsymbol{z}}{\sum_{(\boldsymbol{z}, y) \in \mathcal{S}_{\text{train}}} \mathbb{1}\{y = c\}},$$

where $c = 1, \ldots, C$. Hence, the associated classifier is:

$$h_{1NN} : \mathcal{Z} \to \mathcal{Y},$$
$$\boldsymbol{z} \mapsto \underset{c \in \mathcal{Y}}{\arg \min} \left\{ \frac{1}{2} - \frac{1}{2} \frac{\langle \boldsymbol{k}_c, \boldsymbol{z} \rangle}{||\boldsymbol{k}_c||_2 ||\boldsymbol{z}||_2} \right\},$$

where we use the *cosine dissimilarity* to quantify the distance between features and prototypes.

The accuracy is evaluated for cluster probing as in equation 7, using $h_{1NN}$:

$$\text{CP accuracy} = \frac{1}{|\mathcal{S}_{\text{TEST}}|} \sum_{(\boldsymbol{z}, y) \in \mathcal{S}_{\text{TEST}}} \mathbb{1}\{h_{1NN}(\boldsymbol{z}) = y\}.$$

Intuitively, an high accuracy in the cluster probing means that the features of the same class are well clustered (spherically), while clusters of different classes are nicely separated (always spherically). This is a very desirable property for unsupervised domain adaptation where we need to leverage on the underlying structure, since labels of the target domain are not available.

# B    Simple Class Alignment

The Simple Class Alignment (SCA) method is part of many domain adaptation algorithms like CAN (Kang et al., 2019) and SHOT (Liang et al., 2020). Nevertheless, in literature, the method is never employed alone and its contributions, as part of the overall DA methods, have not been explored extensively.

The algorithm stands out for its simplicity: (1) find a prototype for every class of the source domain, (2) use these prototypes as initialization for K-Means on the unlabeled target domain and finally, (3) use the resulting prototypes as a 1-NN classifier for the target domain, i.e., classify every sample based on the class of the closest prototype. As in previous works, we used cosine dissimilarity (spherical K-Means) to evaluate how close two feature vectors are.

We highlight that this method finds a new classifier for the new domain by aligning the prototypes to the target distribution. Hence, it can be effective with features extracted from a frozen model without fine-tuning. Consequently, SCA is a highly efficient Domain Adaptation method that can achieve results comparable with the state-of-the-art.

But, *how to find the initialization prototypes?* We compare 4 different techniques considering a problem where we have $C$ classes and a fixed feature extractor that can be used to extract, from images, feature vectors in $\mathbb{R}^d$ (we indicate feature vectors with letter $\boldsymbol{z}$).

**From source labels.** Leverage the available source labels to partition the source feature vectors into the sets $\mathcal{C}_1, \ldots, \mathcal{C}_C$, where all features of class $j$ are in $\mathcal{C}_j$. Then, compute the prototype for class $j$ as the average of features of that class:

$$\boldsymbol{w}_j = \frac{1}{|\mathcal{C}_j|} \sum_{\boldsymbol{z} \in \mathcal{C}_j} \boldsymbol{z}. \tag{8}$$

This initialization has been used in CAN (Kang et al., 2019) method.

**From Multinomial Regression weights**. Learn a Multinomial Regression (MR) classifier $h : \boldsymbol{z} \mapsto \sigma(\boldsymbol{W}\boldsymbol{z})$, where $\sigma$ is the Softmax function, by training on the labeled source domain. Then, initialize the prototypes as the rows of the matrix $\boldsymbol{W} \in \mathbb{R}^{C \times d}$. Algorithm 1 shows all the steps of SCA with this initialization.

**From target hard predictions (pseudo-labels)**. Train a MR classifier on the source domain and pseudo-label the target accordingly. Initialize each prototype with the average feature of samples associated to the same pseudo-label. We note that this initialization, followed by spherical K-Means is very similar to the previous one: the only difference is that, in this case, in the first iteration to compute the distances for the assignments the weights of matrix $\boldsymbol{W}$ and features $\boldsymbol{z}$ are not normalized. As we will see, this method do not perform as good as the others.

**From target soft predictions**. Fit a MR classifier on source data and compute prototypes as weighted averages of all samples where weights are given by predicted class probabilities. More precisely, for each feature target vector $z_i$ the output of the MR classifier (after the Softmax) is $\boldsymbol{p}_i = (p_{i1}, \ldots, p_{iC})$ where $p_{ic}$ is the predicted probability of sample $i$ to belong to class $c$. Then, the prototype of class $c$ is evaluated as:

$$\boldsymbol{k}_c = \frac{\sum_i p_{ic} \boldsymbol{z}_i}{\sum_i p_{ic}}. \tag{9}$$

This method has been used in SHOT algorithm.

We remark that these methods can also be applied if a fine-tuning of the feature extractor has been performed or a linear layer is employed in place of the MR classifier.

*Which is the best method to initialize prototypes?* In Tab. 6 we report the average gain on all models of SCA (without fine-tuning), the failure rate, the average gain in case of successful adaptation and the average degradation in case of negative transfer for the 4 types of initialization. The SCA gain/degradation is considered with respect to naive domain generalization. The method that per-

---

**Algorithm 1** SCA from Multinomial Regression weights

---

**Require:**
- Multinomial regressor for source domain with weights $\boldsymbol{W} = (\boldsymbol{w}_1, \boldsymbol{w}_2, \ldots, \boldsymbol{w}_C)^T \in \mathbb{R}^{C \times d}$.
- $\boldsymbol{Z} = (\boldsymbol{z}_1, \boldsymbol{z}_2, \ldots, \boldsymbol{z}_N)^T \in \mathbb{R}^{N \times d}$: unlabeled target samples (features).

$\boldsymbol{1}_d \leftarrow (1, ..., 1)^T$
$\boldsymbol{X} \leftarrow \operatorname{diag}((\boldsymbol{Z} \circ \boldsymbol{Z})\boldsymbol{1}_d)^{-1/2}\boldsymbol{Z}$            ▷ Normalize features to unit length.

**while** not converged **do**

     $\boldsymbol{W} \leftarrow \operatorname{diag}((\boldsymbol{W} \circ \boldsymbol{W})\boldsymbol{1}_d)^{-1/2}\boldsymbol{W}$           ▷ Normalize weights to unit length.

     **for** $i$ in $1, \ldots, N$ **do**          ▷ Assign every sample to the most similar weight vector.
         $c_i \leftarrow \arg\min_{j \in \{1, \ldots, C\}} \langle \boldsymbol{Z}_{i,:}, \boldsymbol{W}_{j,:} \rangle$
     **end for**

     **for** $j$ in $1, \ldots, C$ **do**                         ▷ Update the weight vectors.
         $\boldsymbol{W}_{j,:} \leftarrow \frac{\sum_{i=1}^N \mathbf{1}\{c_i = j\} \boldsymbol{Z}_{i,:}}{\sum_{i=1}^N \mathbf{1}\{c_i = j\}}$
     **end for**

**end while**

**return** $\boldsymbol{W}$

---

forms best is the initialization from the Multinomial Regressor weigths: it is more robust (about 20% failure rate) and the degradation, in case of failure, is modest (1.8%).

The method of hard predictions is the only one that performs poorly, when it succeed it can gain more than other methods (7.2%), but it has a large failure rate and it leads to large performance impairment in case of failure.

**Table 6:** Comparison of the four different SCA initializations. The results are the average of 548 architectures and 74 different domain pairs. The $\Delta$ accuracy is the difference between the target accuracy after SCA and the naive domain generalization of MR classifier trained on source. The failure rate is the percentage of experiments where SCA degrades the naive domain generalization. In the last two rows are reported the accuracy gain and degradation considering the successful and failure adaptation, respectively. These result differ from the ones in the main text, since here we consider the full set of 548 architectures.

|  | Source Labels | MR Weights | Hard Pred. | Soft. Pred. |
|---|---|---|---|---|
| **$\Delta$ Accuracy** | $3.8 \pm 1.2$ | $4.5 \pm 1.1$ | $-0.3 \pm 2.1$ | $3.7 \pm 1.5$ |
| **Failure Rate** | $25.1 \pm 10.2$ | $20.1 \pm 7.2$ | $46.2 \pm 8.1$ | $29.3 \pm 10.2$ |
| **$\Delta$ Accuracy \| Success** | $5.8 \pm 1.0$ | $6.2 \pm 0.9$ | $7.2 \pm 0.1$ | $6.4 \pm 1.0$ |
| **$\Delta$ Accuracy \| Failure** | $-1.8 \pm 0.6$ | $-1.8 \pm 1.0$ | $-15.4 \pm 4.3$ | $-2.4 \pm 1.6$ |

## C ARCHITECTURES

In Fig. 4 we present some plots to help the localization of architecture families on the scatter plots. The architectures of the same family differs for size, e.g., Resnet50 and Resnet101, training procedure (even if the training is always performed on ImageNet or ImageNet21k) or for minor architectural details.

**Subset of 59 models.** For fine-tuning and SHOT experiments we used a subset of 59 models taken from the timm library (with the exception of resnet50_in21k, resnext101_32x8d_in21k,

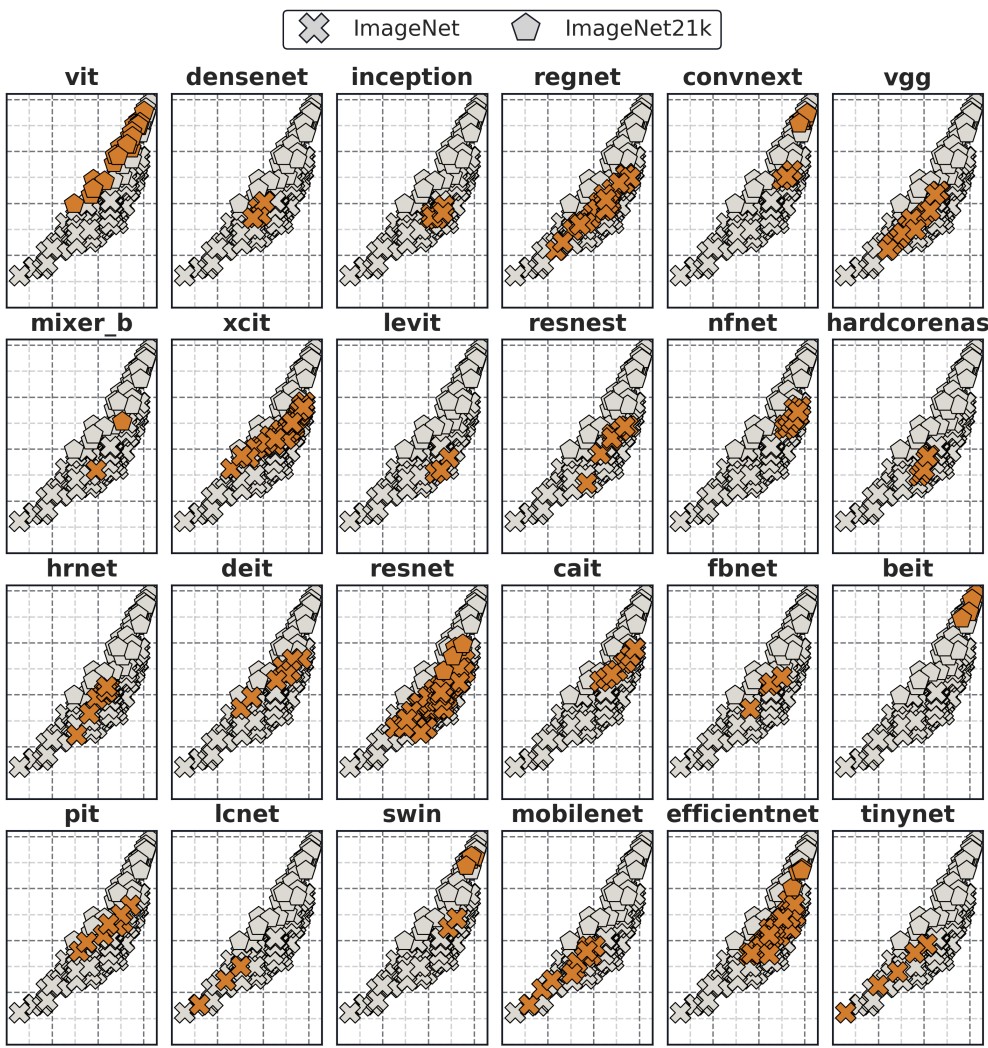

**Figure 4:** Families of architectures: on the x-axis we have the ImageNet top1 accuracy, on the y-axis the accuracy after applying SCA (without fine-tuning) averaged across 74 domain shifts.

densenet161_in21k, wide_resnet50_2_in21k, vgg19_bn_in21k, that were trained by us). The models are:

beit_base_patch16_384, beit_large_patch16_224, beit_large_patch16_384, convnext_base, convnext_base_in22ft1k, convnext_large, convnext_xlarge_384_in22ft1k, convnext_xlarge_in22ft1k, deit_base_distilled_patch16_224, deit_small_distilled_patch16_224, dla169, ecaresnet101d, efficientnetv2_rw_m, efficientnet_b4, efficientnet_el, efficientnet_em, efficientnet_es, fbnetv3_d, ghostnet_100, gluon_resnet50_v1c, gluon_resnext101_32x4d, gmixer_24_224, hrnet_w44, jx_nest_tiny, mixer_b16_224_miil, mixnet_m, pit_ti_224, pit_xs_distilled_224, repvgg_b3g4, resnet152d, resnet18, resnet26, resnet50, selecsls60, swin_base_patch4_window12_384, swin_base_patch4_window7_224, swin_large_patch4_window7_224, tf_efficientnetv2_l_in21ft1k, tf_efficientnetv2_s_in21ft1k, tf_efficientnetv2_xl_in21ft1k, tf_mobilenetv3_small_100, vgg19_bn, visformer_small, vit_base_patch16_224_miil, vit_base_patch8_224, vit_base_r50_s16_384, vit_large_patch16_224, vit_large_patch16_384, vit_large_patch32_384, vit_small_patch16_384, vit_small_patch32_224, vit_tiny_patch16_384, xception65, xception71, resnet50_in21k, resnext101_32x8d_in21k, densenet161_in21k, wide_resnet50_2_in21k, vgg19_bn_in21k.

## D  TECHNICAL DETAILS

For all experiments we used Python 3.7.6 with PyTorch 1.12.1 and CUDA 10.2.

We followed the common procedure used in SF-UDA (see for example SHOT Liang et al. (2020)) with some adjustments to face the large number of experiments and the great variety and diversity of datasets and models.

**Model.** For every pre-trained model we remove the last linear layer (ImageNet classifier). Then, we add a randomly initialized bottleneck followed by a linear layer (classifier for the given task). The bottleneck is composed by a linear layer that maps the features of the backbone to 256 dimensions, followed by a normalization layer and a ReLU activation. The bottleneck normalization layer depends on the backbone, i.e., Batch Normalization if the backbone contains Batch Normalization layers, otherwise it is a Layer Normalization layer.

**Training.** The distributed training is performed on a variable number of GPUs depending on the model size (between 2 and 16 Nvidia V100 16GB) and automatic mixed precision is used by default. The batch normalization statistics are always synchronized between different processes. The global batch size is kept fixed at 64 and, in the cases where the memory of 16 GPUs is not enough, gradient accumulation is used. SGD with Nesterov momentum (0.9) is used for the optimization. The initial learning rate for the classifier and the bottleneck is 0.01, while the initial learning rate of the pre-trained backbone is 0.001. The learning rates are decreased following the same exponential scheduling used in SHOT. During training, the 15% of the training set is kept as validation (used just to determine the stopping iteration). To face the great diversity of dataset sizes we limit the number of optimization steps of each epoch to 100 (in practise we considered one epoch just the interval between two validations and not, as it is usual, an iteration of the full dataset). We fixed the minimum number of epochs at 10 and the maximum number at 100. After each epoch the validation accuracy is evaluated and if there are 5 epochs in a row where the validation accuracy does not increase, the training is stopped and the model weights with the highest validation accuracy are returned. As regularization we used weight decay (0.001) and label smoothing (0.1) on the standard crossentropy loss.

**SHOT.** For the adaptation with SHOT we limited the number of epochs at 15 (always with a maximum of 100 optimization steps) and we used the same hyperparameters of the original paper. We always used distributed training and automatic mixed precision.

**NRC.** For NRC (Yang et al., 2021a) method we used the official code (not distributed) with the official hyperparameters. We added just automatic mixed precision to allow the training of larger models.

**ImageNet21k training.** To train ResNet50, VGG19, Wide-Resnet50, Resnext101 32x8d and DenseNet161 on ImageNet21k we used the official torchvision training script with the default hyperparamters, with the exception of:

- initial learning rate: 0.8
- batch size: 2048
- lr schedule: constant decay every of 0.1 every 15 epochs
- epochs: 50

No augmentations are used and neither a validation set, the final weights are tested on the downstream SF-UDA tasks.

# E    STATISTICAL ANALYSIS OF RESULTS

*Are the pre-training differences that we observed relevant?* Even if, from the scatter plots presented in Fig. 1 it is possible to see, visually, that there is an interesting gap between models pre-trained on ImageNet and ImageNet21k we fit a linear model to understand the benefits of introducing pre-training as independent variable. In particular, for any experiment, first we try to associate the result accuracy that it is possible to obtain with a given backbone $\mathcal{B}$ to its ImageNet top1 accuracy:

$$accuracy(\mathcal{B}) = m \cdot top_1(\mathcal{B}) + q \tag{10}$$

It is possible to find the parameters $m$ and $q$ with the ordinary least squares estimator. Then, we encode the two pre-training into a binary function such that if the backbone is pre-trained on ImageNet $pretrain(\mathcal{B}) = 0$, while if it is pre-trained on ImageNet21k $pretrain(\mathcal{B}) = 1$ and we introduce it into the model. In particular we can consider the following:

- **Different intercept model**

$$accuracy(\mathcal{B}) = m \cdot top_1(\mathcal{B}) + (q + \Delta q \cdot pretrain(\mathcal{B})) \tag{11}$$

- **Different slope model (interaction)**

$$accuracy(\mathcal{B}) = (m + \Delta m \cdot pretrain(\mathcal{B})) \cdot top_1(\mathcal{B}) + q \tag{12}$$

- **Different slope and intercept**

$$accuracy(\mathcal{B}) = (m + \Delta m \cdot pretrain(\mathcal{B})) \cdot top_1(\mathcal{B}) + (q + \Delta q \cdot pretrain(\mathcal{B})) \tag{13}$$

To compare the different models we use the goodness-of-fit ($R^2$) that represents the fraction of explained variance. Given the observed values $(\boldsymbol{x}_1, y_1), \ldots, (\boldsymbol{x}_N, y_N)$ (where $x_i$ is a vector of independent variables, while $y_i$ is the associated dependent variable) and the values predicted by the model $f$: $\hat{y}_1 = f(\boldsymbol{x}_1), \ldots, \hat{y}_N = f(\boldsymbol{x}_N)$ we can compute the mean of the observed data as:

$$\bar{y} = \frac{1}{N} \sum_{i=1}^{N} y_i \tag{14}$$

the *total sum of squares* is defined as:

$$SS_{tot} = \sum_{i=1}^{N} (y_i - \bar{y})^2 \tag{15}$$

while the *residual sum of squares*:

$$SS_{res} = \sum_{i=1}^{N} (y_i - \hat{y}_i)^2 \tag{16}$$

then the coefficient of determination can be defined as:

$$R^2 = 1 - \frac{SS_{res}}{SS_{tot}} \tag{17}$$

Then, even if in our case the difference is marginal, we use the adjusted $R^2$ (indicated as $\bar{R}^2$) to account for the different number of independent variables in the models, that is defined as:

$$\bar{R}^2 = 1 - (1 - R^2) * \frac{N - 1}{N - df} \tag{18}$$

where $df$ are the degrees-of-freedom of the model.

We report all the coefficients of the linear and multi-linear models and the $\bar{R}^2$ values in Tab. 7. An illustration is given in Figs. 5-14. In some cases (for example in SHOT), we get the p-values for the $\Delta m$ and $\Delta q$ parameters higher than the statistical significance of 0.01. In this case, we just need to remove one of the two parameters to make remaining ones all significant. This phenomenon is due to the fact that the two lines, in these experiments, are almost parallel and so, $\Delta m$ is approximately 0 (see Fig. 10) and can be safely be removed from the statistical model without decreasing the goodness-of-fit.

**Table 7:** Coefficients and $\bar{R}^2$ obtained with single and multi linear models. Parameters of the multi-linear model with p-value higher than 0.01 are removed from the model, since their contribution is not statistically significant.

| Experiment | model | m | Δm | Δq | q | $\bar{R}^2$ |
|---|---|---|---|---|---|---|
| LP Gen | single reg. | 0.61 | — | — | 0.29 | 0.74 |
| | multiple reg. | 0.51 | 0.31 | −0.23 | 0.36 | 0.85 |
| CP Gen | single reg. | 0.72 | — | — | 0.16 | 0.76 |
| | multiple reg. | 0.59 | 0.23 | −0.14 | 0.26 | 0.91 |
| LP DGen | single reg. | 1.14 | — | — | −0.41 | 0.81 |
| | multiple reg. | 0.95 | 0.62 | −0.45 | −0.26 | 0.93 |
| CP DGen | single reg. | 1.14 | — | — | −0.41 | 0.75 |
| | multiple reg. | 0.92 | 0.54 | −0.38 | −0.25 | 0.92 |
| SCA | single reg. | 1.08 | — | — | −0.32 | 0.73 |
| | multiple reg. | 0.87 | 0.42 | −0.27 | −0.16 | 0.90 |
| SHOT | single reg. | 1.62 | — | — | −0.71 | 0.79 |
| | multiple reg. | 1.21 | — | 0.07 | −0.40 | 0.89 |
| FT Gen | single reg. | 0.62 | — | — | 0.35 | 0.80 |
| | multiple reg. | 0.48 | — | 0.02 | 0.46 | 0.88 |
| FT DGen | single reg. | 1.68 | — | — | −0.82 | 0.70 |
| | multiple reg. | 1.16 | — | 0.09 | −0.43 | 0.82 |
| FT+SCA | single reg. | 1.62 | — | — | −0.72 | 0.67 |
| | multiple reg. | 1.10 | — | 0.08 | −0.33 | 0.80 |
| FT+SHOT | single reg. | 1.51 | — | — | −0.59 | 0.84 |
| | multiple reg. | 1.15 | — | 0.06 | −0.32 | 0.93 |

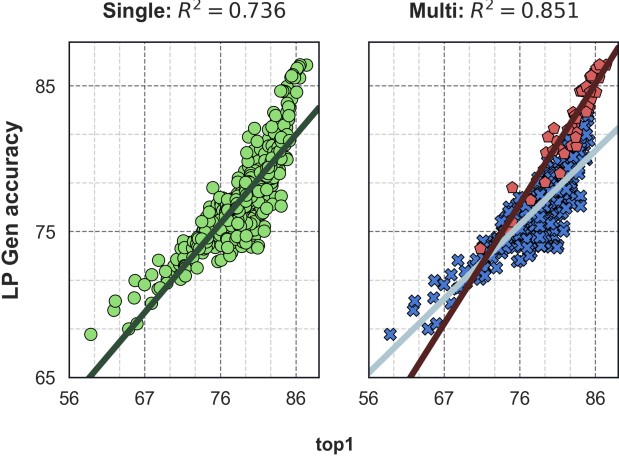

**Figure 5:** Average Linear Probing generalization accuracy (over 23 domains).

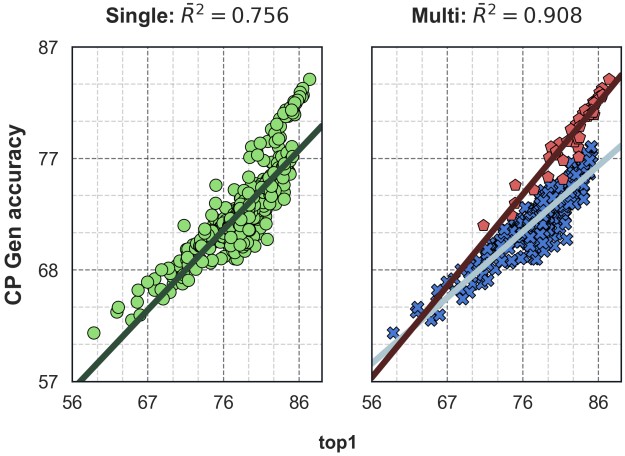

**Figure 6:** Average Cluster Probing generalization accuracy (over 23 domains).

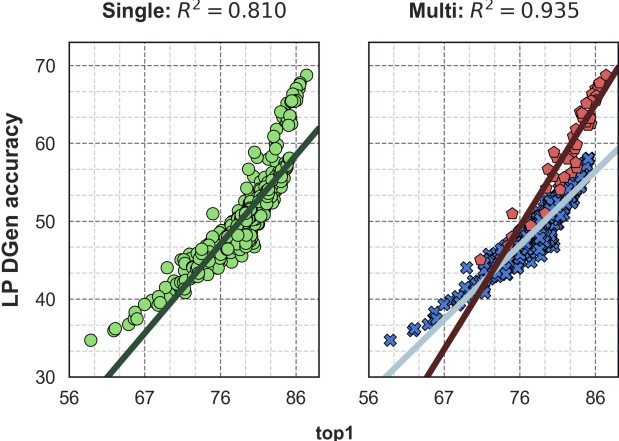

**Figure 7:** Average Linear Probing domain generalization accuracy (over 74 domain shifts).

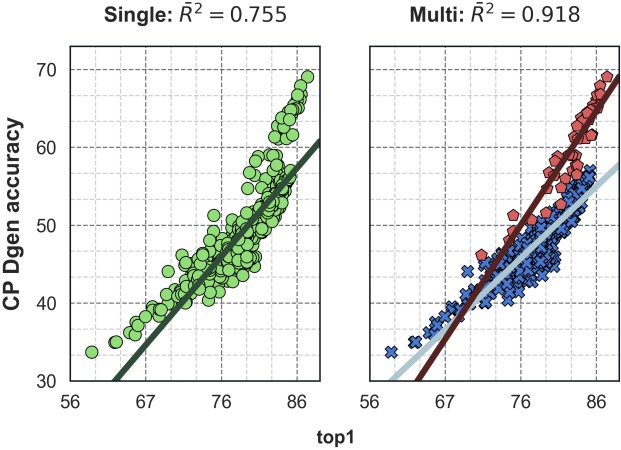

**Figure 8:** Average Cluster Probing domain generalization accuracy (over 74 domain shifts).

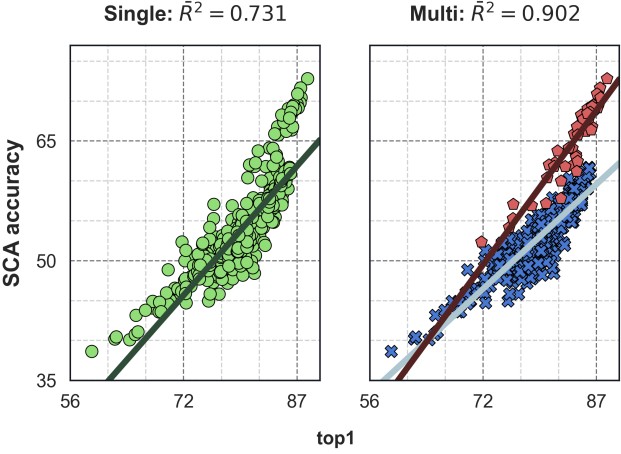

**Figure 9:** Average accuracy (over 74 domain shifts) of models after applying SCA (without fine-tuning).

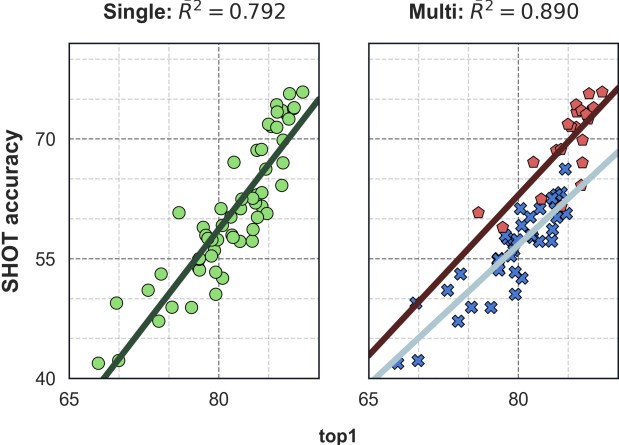

**Figure 10:** Average accuracy (over 74 domain shifts) of models after applying SHOT (without fine-tuning).

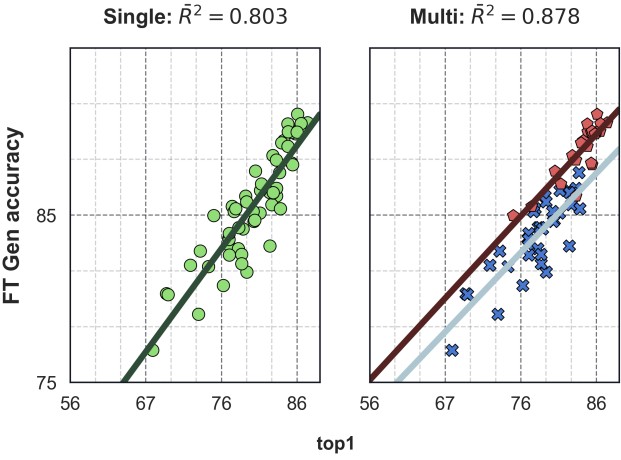

**Figure 11:** Average generalization accuracy (over 23 domains) of models after applying the FT.

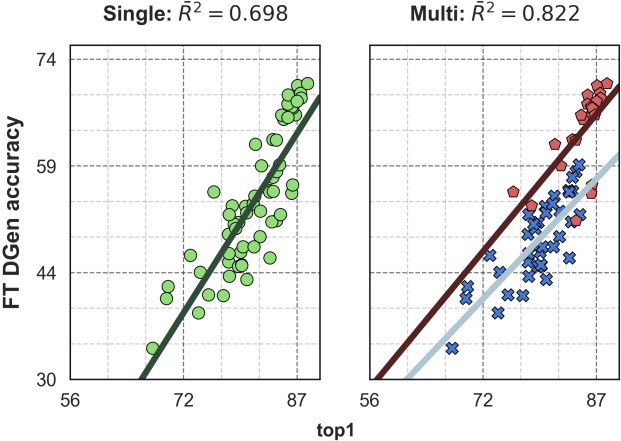

**Figure 12:** Average domain generalization accuracy (over 74 domain shifts) of models after applying the FT.

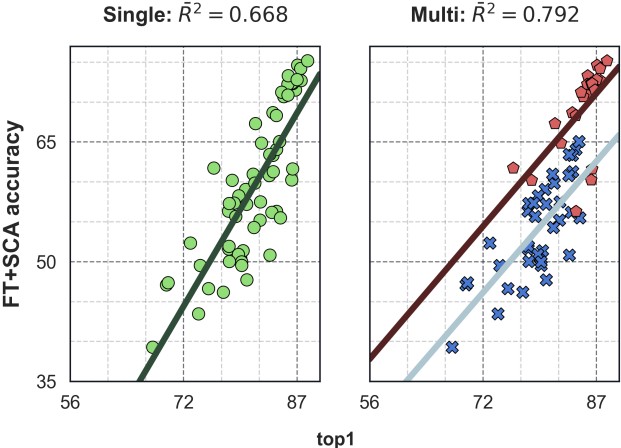

**Figure 13:** Average accuracy (over 74 domain shifts) of models after applying the FT and SCA.

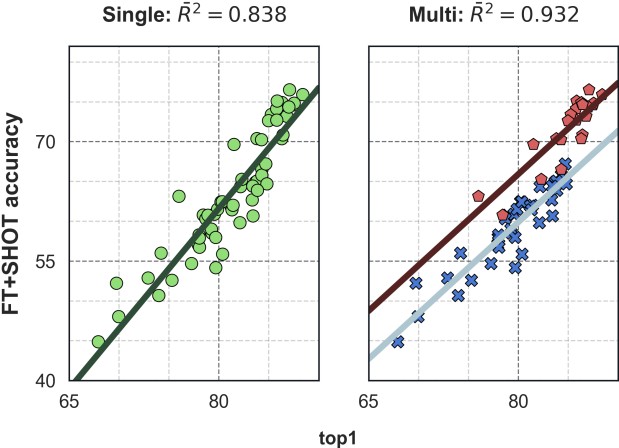

**Figure 14:** Average accuracy (over 74 domain shifts) of models after applying the FT and SHOT. In this case the two lines of the statistical multi-linear model are parallel ($\Delta m$ was not statistically

# F Comparison of Domain Generalization and SCA with semantic pre-training

We further investigate the role of pre-training strategy. Tab. 8 contrasts the domain generalization accuracy for miil (Ridnik et al., 2021) and naive pre-training on different domain generalization datasets. Tab. 9 shows results for different initializations of ResNet50 on Office31.

**Table 8:** Comparison of results (accuracies %) of semantic pre-training (miil) of Ridnik et al. (2021) and naive pre-training of ResNet50, for linear probing domain generalization and SCA. The results are avaraged across the domains of the datasets.

| Experiment | Pre-train | O31 | Visda | O.Home | Adapt. | I-CLEF | D.Net |
|---|---|---|---|---|---|---|---|
| LP DGen | naive[†] | 86.6 | 63.1 | 68.5 | 59.1 | 79.8 | 19.9 |
| | miil | 88.8 | 67.3 | 71.4 | 68.2 | 81.9 | 25.3 |
| SCA | naive[†] | 90.3 | 80.7 | 74.4 | 73.8 | 85.1 | 23.6 |
| | miil | 90.4 | 79.8 | 72.2 | 73.3 | 84.8 | 21.3 |

[†] Performed by us.

**Table 9:** Comparison of results (accuracies %) on Office31 benchmark using different pre-trained weights of Resnet50. We included the following pre-training: **timm** (Wightman, 2019), the semantic training of Ridnik et al. (2021) (**miil**) and our naive pretrain. Office31 domains are **A**mazon, **W**ebcam and **D**SLR and the experiments are indicated with **source → target**. For these experiments the official code (not distributed and without Automatic Mixed Precision) of SHOT (Liang et al., 2020) and NRC (Yang et al., 2021a) have been used, so the results may be slightly different from the ones reproduced with our implementation.

| | | | | Office31 (ResNet50) | | | | | |
|---|---|---|---|---|---|---|---|---|---|
| **Method** | **Init** | **IN21k** | **A → W** | **A → D** | **W → A** | **W → D** | **D → A** | **D → W** | **Avg** |
| LP DGen | timm | | 77.5 | 78.9 | 68.3 | 97.8 | 64.6 | 96.6 | 80.6 |
| | miil | ✓ | 89.4 | 89.1 | 78.3 | 99.4 | 78.7 | 98.1 | 88.8 |
| | naive | ✓ | 86.2 | 83.3 | 77.3 | 99.4 | 76.4 | 97.1 | 86.6 |
| SCA | timm | | 89.2 | 92.5 | 73.4 | 97.2 | 73.9 | 90.1 | 86.0 |
| | miil | ✓ | 93.8 | 97.4 | 79.0 | 98.2 | 78.8 | 95.2 | 90.4 |
| | naive | ✓ | 93.3 | 94.6 | 80.1 | 98.2 | 80.0 | 95.7 | 90.3 |
| FT+SHOT | timm | | 89.3 | 87.1 | 68.9 | 99.8 | 67.1 | 98.1 | 85.5 |
| | miil | ✓ | 93.2 | 92.4 | 75.3 | 99.8 | 76.8 | 98.8 | 89.4 |
| | naive | ✓ | 91.1 | 94.0 | 75.3 | 99.6 | 73.5 | 98.8 | 88.7 |
| FT+NRC | timm | | 94.1 | 93.0 | 72.2 | 99.8 | 69.8 | 98.2 | 87.7 |
| | miil | ✓ | 96.0 | 89.6 | 76.7 | 99.6 | 75.0 | 98.1 | 89.2 |
| | naive | ✓ | 92.6 | 96.0 | 80.4 | 99.8 | 78.5 | 98.4 | 90.9 |

# G   BATCH NORMALIZATION VS LAYER NORMALIZATION

Fixing a source-target domain pair, we compute the average domain generalization improvement/degradation over the different architectures. In particular, we split all 59 architectures considered into 4 groups based on normalization layers and pre-training: BN+IN (24 models), BN+IN21k (8 models), LN+IN (8 models) and LN+IN21k (19 models). In Fig. 15 we show the average improvement/degradation for each domain pair and for every model group. As it is possible to see, there are some domain pairs where the models with BN degrades a lot the domain generalization, while all models with LN are more stable.

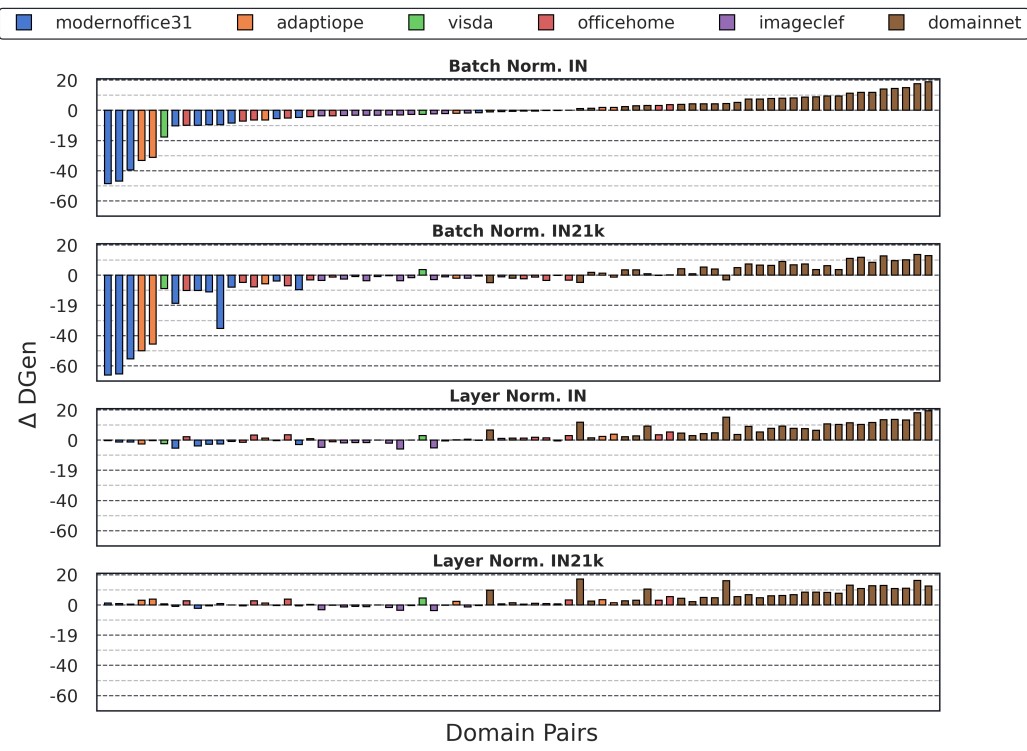

**Figure 15:** Each bar represent a different domain-shift experiment, i.e., a source and a target. Bars height are the average gain/degradation of domain generalization (always w.r.t. the baseline LP domain generalization) for the given domain pair.

In Tab. 10 are reported the numerical results for the domain pairs with the highest degradation caused by the fine-tuning with Batch Normalization layers. In particular, the domain pairs correspond to the 10 left-most bars of Fig. 15

In Tab. 11 we report the average degradation in case of failure on all domain pairs for different model groups. Also here it is possible to see that the models with Batch Normalization are affected from high degradation of the domain generalization with the fine-tuning.

**Table 10:** Domain pairs with the highest fine-tuning domain generalization degradation caused by Bacth Normalization layers.

| dataset | source | target | bn_in1k | bn_in21k | ln_in1k | ln_in21k |
|---|---|---|---|---|---|---|
| modernoffice31 | synthetic | webcam | -48.6 | -66.1 | -0.3 | 1.2 |
| modernoffice31 | synthetic | dslr | -46.8 | -65.4 | -1.3 | 1.0 |
| modernoffice31 | synthetic | amazon | -39.5 | -55.4 | -1.3 | 0.6 |
| adaptiope | synthetic | real_life | -33.1 | -50.0 | -2.6 | 3.2 |
| adaptiope | synthetic | product_images | -31.2 | -45.7 | -0.3 | 3.9 |
| visda | train | validation | -17.6 | -8.7 | -2.4 | 0.8 |
| modernoffice31 | dslr | synthetic | -10.3 | -18.5 | -5.3 | -0.9 |
| officehome | Clipart | Art | -10.0 | -10.2 | 2.3 | 2.7 |
| modernoffice31 | webcam | synthetic | -10.0 | -10.0 | -3.9 | -2.2 |
| modernoffice31 | webcam | amazon | -9.6 | -11.0 | -2.7 | -0.5 |

**Table 11:** Average degradation (%) in case of failure (averaged across 74 domain pairs). Every row select a certain type of model to make a full comparison. Between parethesis in reported the number of model in that group.

| Models | FT DGen | SCA | SHOT | FT+SCA | FT+SHOT |
|---|---|---|---|---|---|
| **All** (59) | -6.67 $\pm$ 4.55 | -1.55 $\pm$ 0.67 | -3.11 $\pm$ 2.76 | -7.88 $\pm$ 6.75 | -1.58 $\pm$ 2.04 |
| **IN** (32) | -8.05 $\pm$ 3.57 | -1.79 $\pm$ 0.68 | -2.66 $\pm$ 1.38 | -8.79 $\pm$ 5.75 | -1.36 $\pm$ 0.91 |
| **IN21K** (27) | -5.02 $\pm$ 5.07 | -1.27 $\pm$ 0.54 | -3.64 $\pm$ 3.76 | -6.65 $\pm$ 7.86 | -1.89 $\pm$ 2.97 |
| **BN** (32) | -10.51 $\pm$ 2.22 | -1.92 $\pm$ 0.63 | -2.83 $\pm$ 1.34 | -12.72 $\pm$ 4.89 | -1.82 $\pm$ 2.46 |
| **LN** (27) | -2.10 $\pm$ 0.65 | -1.12 $\pm$ 0.40 | -3.43 $\pm$ 3.82 | -1.41 $\pm$ 0.77 | -1.27 $\pm$ 1.26 |
| **IN+BN** (24) | -9.82 $\pm$ 2.01 | -1.95 $\pm$ 0.70 | -2.56 $\pm$ 0.98 | -11.26 $\pm$ 4.37 | -1.27 $\pm$ 0.72 |
| **IN+LN** (8) | -2.75 $\pm$ 0.63 | -1.32 $\pm$ 0.34 | -2.96 $\pm$ 2.26 | -1.39 $\pm$ 0.57 | -1.60 $\pm$ 1.33 |
| **IN21k+BN** (8) | -12.60 $\pm$ 1.40 | -1.82 $\pm$ 0.40 | -3.65 $\pm$ 1.94 | -17.10 $\pm$ 3.75 | -3.31 $\pm$ 4.47 |
| **IN21k+LN** (19) | -1.83 $\pm$ 0.43 | -1.03 $\pm$ 0.40 | -3.63 $\pm$ 4.35 | -1.42 $\pm$ 0.87 | -1.08 $\pm$ 1.22 |

## H    Computational times.

In Tab. 12 we report the average number of seconds to perform SCA, one epoch of fine-tuning and one epoch of SHOT adaptation for Office31 and Visda datasets (averaged across all different domains). We remark that SCA is performed just once, while the fine-tuning (in our experiments) uses between 15 (10 + 5, see technical details in App. D) and 100 epochs, while SHOT uses always 15 epochs.

| Dataset | Model | FT | SCA | SHOT | DANN |
|---------|-------|----|----|------|------|
| Office31 | ResNet50 | 3.4 | 1.7 | 6.7 | 7.0 |
|          | ViT Base | 5.7 | 4.5 | 14.8 | 10.8 |
| Visda    | ResNet50 | 211.1 | 92.1 | 425.3 | 404.1 |
|          | ViT Base | 381.6 | 327.1 | 1068.5 | 790.3 |

**Table 12:** Time required (in seconds) for fine-tuning and adaptation. The times are averaged on the domains of the used dataset. For fine-tuning and SHOT the time reported refers to one epoch. We remark that according to standard training, SHOT requires 15 epochs to adapt while DANN (Ganin et al., 2016) requires 100 epochs. Models are in mixed precision training with batch size 64 split into 2 × NVIDIA RTX 2070 SUPER on a single machine with AMD Ryzen 3900x.

## I    Additional results: best performing models.

In Tab. 13 we provide domain generalization results on large architectures on different DA datasets when fine-tuning varying on the fine-tuning choice.

**Table 13:** Domain generalization accuracy for large backbones with different fine-tuning decisions. Both the fine-tuning and non fine-tuning cases are considered. For each architecture, the accuracy change is the average over the domains of the considered dataset.

| Model | Method | FT | Office31 | Visda | OfficeHome | Adapt. | Mod.Office31 | Image-CLEF | DomainNet |
|-------|--------|----|----------|-------|------------|--------|--------------|------------|-----------|
| BEIT L 384 | SCA | ✗ | 94.8 | 85.1 | 89.0 | 93.1 | 97.4 | 88.6 | 45.3 |
|            |     | ✓ | 94.2 | 79.6 | 88.9 | 92.7 | 95.5 | 86.4 | 45.0 |
|            | SHOT | ✗ | 95.5 | 89.2 | 92.2 | 95.8 | 96.5 | 88.3 | 45.8 |
|            |      | ✓ | 94.6 | 90.7 | 90.1 | 95.1 | 96.6 | 89.6 | 45.1 |
| SWIN L 224 | SCA | ✗ | 95.0 | 83.4 | 84.9 | 88.8 | 96.3 | 88.3 | 37.9 |
|            |     | ✓ | 94.4 | 83.0 | 86.4 | 88.2 | 93.3 | 86.0 | 49.1 |
|            | SHOT | ✗ | 94.8 | 88.5 | 90.6 | 91.5 | 96.4 | 88.5 | 47.1 |
|            |      | ✓ | 95.0 | 88.9 | 89.3 | 91.9 | 96.0 | 88.0 | 51.2 |
| VIT L 384 | SCA | ✗ | 95.3 | 82.6 | 88.0 | 89.3 | 96.4 | 88.2 | 42.4 |
|           |     | ✓ | 94.0 | 83.5 | 88.6 | 89.5 | 94.6 | 86.7 | 52.6 |
|           | SHOT | ✗ | 94.9 | 89.0 | 91.5 | 93.9 | 95.4 | 89.2 | 51.5 |
|           |      | ✓ | 93.9 | 83.9 | 91.3 | 92.4 | 95.8 | 89.5 | 54.1 |
| ConvNext XL 384 | SCA | ✗ | 95.5 | 84.9 | 86.9 | 90.0 | 96.5 | 88.8 | 41.2 |
|                 |     | ✓ | 94.8 | 84.1 | 87.4 | 90.8 | 93.8 | 83.2 | 49.8 |
|                 | SHOT | ✗ | 94.0 | 89.2 | 90.3 | 93.9 | 94.0 | 89.0 | 48.5 |
|                 |      | ✓ | 95.1 | 88.1 | 88.8 | 93.3 | 92.2 | 89.4 | 51.0 |

## J    Self-supervised pre-training

In all previous experiments we considered only supervised pre-training. However, recently, the community shows a great interest on self-supervised learning (SSL) techniques, which proved to achieve remarkable performance. Thus in Tab. 14, we present the results of some experiments on ResNet50 trained with three different SSL methods, namely, DINO (Caron et al., 2021), MOCO v1 (He et al., 2020) and MOCO v2 (Chen et al., 2020). As it can be noticed, after performing a fine-tuning on the source domain (and eventually applying SCA) MOCO v2 performs very similarly (on average) with a supervised pre-training on ImageNet, while DINO and MOCO v1 achieve a lower accuracy. When FT+SHOT is applied MOCO v1 still underperforms other methods, while DINO and MOCO v2 achieve similar and promising results which, however, are still marginally below the performance of the supervised pre-training, which gets the best accuracy (66.1%).

From our findings and extensive experiments reported in the main text, we know that the fine-tuning on the source domain of a model with Batch Normalization layers (like ResNet50) can lead to

large drop in performance in some scenarios and the right way to adapt these models after the fine-tuning is to adapt the feature extractor (with SHOT, for example). Moreover, although the results are promising for SSL pre-training and even if MOCO-v2 performed similarly to supervised pre-training on FT and FT+SCA, the drop in accuracy for FT+SHOT highlights that SSL pre-training still underperforms supervised pre-training (on ImageNet1k) on SF-UDA, when the right adaptation choices are made.

**Table 14:** Target accuracy (%) using different initialization weights of ResNet50 after performing the following experiments: fine-tuning on the source domain (FT), FT+SCA and FT+SHOT.

| Experiment | Pre-train | MO31 | Visda | O.Home | Adapt. | I-CLEF | D.Net | Avg |
|---|---|---|---|---|---|---|---|---|
| FT | DINO | 42.7 | 45.6 | 45.7 | 30.9 | 71.3 | 19.7 | 42.7 |
| | MOCO v1 | 38.5 | 44.6 | 41.7 | 28.6 | 66.2 | 20.1 | 40.0 |
| | MOCO v2 | 54.7 | **54.2** | 55.1 | 39.5 | **76.9** | **25.5** | **51.0** |
| | Supervised (IN1k) | **56.4** | 49.6 | **55.5** | **43.3** | 75.9 | 21.6 | 50.4 |
| FT+SCA | DINO | 50.8 | 54.5 | 49.8 | 35.9 | 75.0 | 21.9 | 48.0 |
| | MOCO v1 | 49.5 | 52.3 | 43.9 | 34.3 | 70.4 | 23.9 | 45.7 |
| | MOCO v2 | **63.4** | **56.9** | 55.8 | 45.3 | **79.5** | **28.0** | 54.8 |
| | Supervised (IN1k) | 62.7 | 55.5 | **59.4** | **49.6** | 79.3 | 23.6 | **55.0** |
| FT+SHOT | DINO | 71.8 | 67.3 | 58.0 | 55.4 | 77.5 | 24.1 | 59.0 |
| | MOCO v1 | 56.6 | 58.3 | 52.9 | 37.5 | 71.7 | 19.2 | 49.4 |
| | MOCO v2 | 73.2 | 65.7 | 66.9 | 51.9 | 81.0 | 26.2 | 60.8 |
| | Supervised (IN1k) | **80.6** | **71.6** | **68.9** | **66.6** | **81.6** | **27.3** | **66.1** |

In Table 15 we report the results on 4 datasets to compare the DINO SSL pre-training to a supervised pre-training on ImageNet21k on a larger architecture that uses Layer Normalization (VIT Base). The results confirm that, in most cases, the gap between SSL and supervised pre-training is still quite large. We also report some instabilities of SHOT using the DINO pre-training (for example on VisDA dataset) that very rarely happened with supervised pre-training.

**Table 15:** Target accuracy (%) using different initialization weights of VIT Base after performing the following experiments: fine-tuning on the source domain (FT), FT+SCA and FT+SHOT.

| Experiment | Pre-train | O31 | MO31 | Visda | O.Home |
|---|---|---|---|---|---|
| FT | DINO | 80.7 | 72.4 | 65.3 | 63.6 |
| | Supervised (IN21k) | **89.7** | **86.9** | **77.9** | **79.9** |
| FT+SCA | DINO | 85.2 | 81.1 | 73.6 | 67.9 |
| | Supervised (IN21k) | **92.7** | **91.9** | **82.6** | **82.5** |
| FT+SHOT | DINO | 87.5 | 75.8 | 56.9 | 74.2 |
| | Supervised (IN21k) | **93.7** | **93.9** | **87.5** | **84.4** |

