# OpenReview forum: "Key Design Choices for Double-transfer in Source-free Unsupervised Domain Adaptation"
_ICLR.cc/2023/Conference — Submitted to ICLR 2023_

### Official Review · Reviewer_R8P1 · 2022-10-22

**Confidence:** 5
**Correctness:** 4
**Technical Novelty And Significance:** 3
**Empirical Novelty And Significance:** 3
**Recommendation:** 6

**Clarity, Quality, Novelty And Reproducibility:**

This paper is very written and conducts an extensive study on the source data-free unsupervised domain adaptation problem. Such motivation is interesting and important.

**Strength And Weaknesses:**

Pros

1. this studied problem is interesting and receives increasing attention in the transfer learning field, making a large-scale empirical study valuable

2. the empirical analysis includes more than 500 architectures (including both CNNs and Vision Transformers), 6 datasets for domain adaptation, which is very extensive

Cons

1. the selected source-free domain adaptation (SFDA) methods are relatively less, only a classifier adjustment method (SCA) and a feature encoder fine-tuning method (SHOT) is validated throughout the study. More SFDA methods like Li et al, CVPR 2020 could be used in an extensive study.

2. generally speaking, the observations would not inspire more ideas in the SFDA topic as most conclusions have already been known.

3. to study SFDA, these choices (the pre-training dataset, the pre-trained network, the normalization strategy) are vital but the authors may focus more on the adaptation strategy, like whether to recover the source data and the robustness to different label distributions (e.g., open-set, partial-set)

Some missing references:

1. Liang, Jian, et al. "Source data-absent unsupervised domain adaptation through hypothesis transfer and labeling transfer." IEEE Transactions on Pattern Analysis and Machine Intelligence (2021).

2. Huang, Jiaxing, et al. "Model adaptation: Historical contrastive learning for unsupervised domain adaptation without source data." Advances in Neural Information Processing Systems 34 (2021): 3635-3649.

3. Kurmi, Vinod K., Venkatesh K. Subramanian, and Vinay P. Namboodiri. "Domain impression: A source data free domain adaptation method." Proceedings of the IEEE/CVF Winter Conference on Applications of Computer Vision. 2021.

4. Xia, Haifeng, Handong Zhao, and Zhengming Ding. "Adaptive adversarial network for source-free domain adaptation." Proceedings of the IEEE/CVF International Conference on Computer Vision. 2021.

Typos:
siCP -> CP in section 3.1

**Summary Of The Paper:**

This paper studies the source-free unsupervised domain adaptation problem and analyzes the impact of several design choices (e.g., the pre-training dataset, the pre-trained network, the normalization strategy, and the adaptation strategy) through a large-scale empirical study.

**Summary Of The Review:**

This paper conducts an extensive study on the source data-free unsupervised domain adaptation problem, which is interesting and important in this field. However, some more studies are required to make this study more valuable.

---

> ### Author Response · Authors · 2022-11-14
> **Answers to Reviewer R8P1 comments. Part 1 of 2.**
>
> > this studied problem is interesting and receives increasing attention in the transfer learning field, making a large-scale empirical study valuable
>
> We acknowledge the Reviewer’s appreciation of the relevance and good timing of our work in current DA research. Moreover, they recognize the value of our large-scale experimentation, which provides sound empirical evidence for our findings.
>
>
> > the empirical analysis includes more than 500 architectures (including both CNNs and Vision Transformers), 6 datasets for domain adaptation, which is very extensive
>
> Although facing constraints on computational resources and space limitations, we provide extensive experiments on backbones, fine-tuning, and normalization layers choice. It turns out they have a surprisingly large and consistent impact on SF-UDA performance.
>
>
> > the selected source-free domain adaptation (SFDA) methods are relatively less, only a classifier adjustment method (SCA) and a feature encoder fine-tuning method (SHOT) is validated throughout the study. More SFDA methods like Li et al, CVPR 2020 could be used in an extensive study.
>
> Regarding evaluated methods, we agree with the Reviewer that validating more methods would further support our claims and we are indeed considering it for future work. Our decision to focus on SHOT and SCA has three main motivations:
> - Conceptually, SHOT and SCA are complementary methods which broadly represent the approaches available in the current literature. SHOT builds on the feature extractor fine-tuning to adapt the source-trained model, i.e., the classifier is fixed and there is a domain-specific feature encoder. In contrast, SCA leverages a shared feature extractor across domains and aligns the classifier.
> - SHOT is a foundational method in the DA literature which inspired many subsequent methods (Taufique et al., 2021, Yang et al., 2021a, Yang et al., 2021b, Qiu et al., 2021)
> - SCA neatly demonstrates how effective the key design choices highlighted in the present work can be: under careful adaptation, an extremely simple method adapting the classifier only is able to compare favorably with the state of the art.
>
> We remark that this work aims to evaluate different architectures, pre-training alternatives, and normalization layer choices for SF-UDA. We demonstrate the great extent to which these design choices are able to affect the adaptation performance.

---

> ### Author Response · Authors · 2022-11-17
> **Answers to Reviewer R8P1 comments. Part 2 of 2.**
>
> > generally speaking, the observations would not inspire more ideas in the SFDA topic as most conclusions have already been known.
>
> We respectfully disagree with the Reviewer’s concern. First, the proposed evaluation methodology might inspire future analysis of the current state of the art: the double-transfer highlights the two transfers we address while performing domain adaptation and the failure rate analysis may help push research towards more robust algorithms. We agree that part of the results experimentally backs up _intuitions_ that were at least partly mentioned in the literature. Nevertheless, to the best of our knowledge, other findings in our work are entirely new: 1) the improved robustness of layer-norm for SF-UDA, and 2) the critical choice of whether to fine-tune or not during the first transfer.
>
> Furthermore, we highlight the impact that the proposed best practices have on a practical setting where a practitioner addresses an SF-UDA problem in a real-world application: (i) the choice of the architecture should be leaning towards the best performing architectures on ImageNet-1K, possibly pretrained on IN21K and with LN, (ii) fine-tuning the model on the source domain is, on average, the best choice, and (iii) applying SF-UDA methods like SHOT (to have better target accuracy) or SCA (to have a fast adaptation while avoiding further modification of the feature extractor) is quite reliable and rarely fails. We stress that selecting the best architectures and SCA can yield very similar performances to state-of-the-art UDA algorithms, while being many orders of magnitude faster (and way simpler).
>
> > to study SFDA, these choices (the pre-training dataset, the pre-trained network, the normalization strategy) are vital but the authors may focus more on the adaptation strategy, like whether to recover the source data and the robustness to different label distributions (e.g., open-set, partial-set)
>
> We thank the Reviewer for this insightful question. We agree that an extensive analysis covering domain adaptation strategies would complement our results. We are considering it for future developments of the project. Still, we stress that the results reported in this work are orthogonal to the adaptation strategy. As mentioned in the question, such implementation details are vital in an application setting to achieve the highest performance given an existing strategy.
>
> Specifically, we would like to highlight the extent of the impact of these choices. We refer to Table 2, in SHOT+FT-IN21K, moving from the benchmark ResNet50-W to a ConvNext B yields an 8.7% accuracy gain. On average, pre-training ConvNext B on ImageNet-21K boosts performances to 72.7% with respect to ImageNet-1K’s 65.1%. Similarly, an impressive drop in robustness is observed on average when considering BN vs. LN: i.e., SHOT-IN21K with BN has a failure rate of 16.89% in contrast to the 5.48% of LayerNorm, see Table 4.
>
>
>
> We thank the Reviewer for having pointed out the missing references, we added them in the revised draft in Section 2.
>
> Note that we anonymously released the experimental code and full results, which are available for download at: [Code](https://drive.google.com/file/d/14cXcFPCuS4dmD5y3I-dJNH94ETlVSxPc/view?usp=sharing)
>
> Based on the further clarifications, we would ask the Reviewer to consider increasing the score for our paper.
> We are open to discussion and glad to provide further elucidations.
>
> **References**
>
> Qiu, Zhen, et al. "Source-free domain adaptation via avatar prototype generation and adaptation." International Joint Conference on Artificial Intelligence(2021).
>
> Taufique, Abu Md Niamul, Chowdhury Sadman Jahan, and Andreas Savakis. "Conda: Continual unsupervised domain adaptation." arXiv preprint arXiv:2103.11056 (2021).
>
> Yang, Shiqi, et al. "Exploiting the intrinsic neighborhood structure for source-free domain adaptation." Advances in Neural Information Processing Systems 34 (2021): 29393-29405.
>
> Yang, Shiqi, et al. "Generalized source-free domain adaptation." Proceedings of the IEEE/CVF International Conference on Computer Vision. 2021b.

---

> > ### Comment · Reviewer_R8P1 · 2022-11-17
> > **Response to the authors' rebuttal**
> >
> > Thanks for the response. After reading the discussion between the authors and the other reviewers, I insist on the initial score (6) as this study currently could not offer some impressive or insightful findings for this field. I sincerely hope the authors could investigate more adaptation methods and more adaptation scenarios and digest some interesting findings in the future.

---

### Official Review · Reviewer_cjNK · 2022-10-24

**Confidence:** 4
**Correctness:** 3
**Technical Novelty And Significance:** 3
**Empirical Novelty And Significance:** 2
**Recommendation:** 5

**Clarity, Quality, Novelty And Reproducibility:**

<Clarity>

- Some terminology is confusing.
	- Does "DGen" mean evaluation of a target model under the domain shift setting? This terminology is quite confusing. (it seems to indicate a certain domain generalization method.)
	- Why does "Generalization LP" perform best in Fig. 1? Is it showing the accuracy at the source domain? If so, this is also confusing, because the other subfigures show the accuracy at the target domain.
	- "DGen -> X" in Fig. 2 is also confusing. It sounds like training the model with a certain domain generalization method at the source domain and then conducting X to adapt the model to the target domain (but it does not mean such a thing).

<Quality>

The experimental setup seems to be reasonable, and the analysis is rigorously based on the results of large-scale experiments.

<Novelty>

Evaluating SFDA methods in double-transfer setting is somewhat new and should be important for practical applications. However, some findings in this study are closely related to those shown in the literature as stated in <weakness>.

<Reproducibility>

The authors state "Experimental data and code will be released upon acceptance."


**Strength And Weaknesses:**



<strength>

- The double-transfer setting studied in this paper seems to be quite practical but has not been well explored in the literature. I love the motivation of this work.
- The experiments cover a very diverse set of datasets as well as model architectures and are done on a large scale.

<weakness>

- Not so surprising results. Some findings in this study are closely related to those shown in the literature; for example, using a large-scale dataset is beneficial in pre-training [Dosovitskiy et al., 2020], and BN has a crucial role when conducting domain adaptation [Li et al., 2017].
- In the experiments, only supervised pre-training is considered. Since self-supervised representation learning has recently become a popular way to pre-train the model, it would be great if it is included in the comparison to reveal the best strategy of pre-training in double-transfer setting.


**Summary Of The Paper:**

This paper reports the performance of SFDA methods under the new setting, called double-transfer, via a large-scale empirical evaluation. The double-transfer represents two-stage transfer learning setting in which the pre-trained model is firstly fine-tuned with a source domain dataset and is then adapted to the target domain. Under this setting, the results shown in this paper reveal several critical factors to achieve better performance after the adaptation.

**Summary Of The Review:**

This paper reports some interesting findings obtained through the empirical studies with a new setting called double-transfer. However, I have several concerns on clarity and novelty as stated above, which makes my score conservative. I vote for "weak reject."

---

> ### Author Response · Authors · 2022-11-14
> **Answers to Reviewer cjNK comments. Part 1 of 2.**
>
> > The double-transfer setting studied in this paper seems to be quite practical but has not been well explored in the literature. I love the motivation of this work.
>
> We thank the Reviewer for the encouraging comments on the motivation of the work. The double transfer has indeed been overlooked in the literature. Analysis of the two transfers involved in DA are critical for advancing the current state of the art and making progress in practical settings.
>
>
> > The experiments cover a very diverse set of datasets as well as model architectures and are done on a large scale.
>
> We believe that extensive experiments are necessary for providing sound empirical results and supporting the proposed claims and best practices, especially in a complex scenario such as SF-UDA.
>
>
> > Not so surprising results. Some findings in this study are closely related to those shown in the literature; for example, using a large-scale dataset is beneficial in pre-training [Dosovitskiy et al., 2020], and BN has a crucial role when conducting domain adaptation [Li et al., 2017].
>
> We agree that our presented results also corroborate previous findings in the literature. However, we extensively demonstrate those findings in a different context, in which we have no guarantees. (Dosovitskiy et al., 2020) shows the benefit of large pre-training for vision transformers in a classical supervised learning setting (i.e., single transfer from pretraining to source domain). In contrast, we reveal that large-scale pre-training data benefits SF-UDA and demonstrate how the domain adaptation performances are related to it: a bilinear model accounting for ImageNet1k accuracy and pre-training data accurately predicts the average SF-UDA performance. The presented results highlight the pivotal role of Batch Normalization for Domain adaptation and yield stronger evidence for previous findings (Li et al., 2017) with large-scale experiments: many distinct models and diverse datasets. Moreover, we shed light on failure modes of BN when normalization statistics of the source data are distant from the target ones and fine-tuning is employed. From a robustness perspective, our analysis indicates that BN architectures deteriorate their performances on average. Layer Norm instead proves more reliable.
>
>
> > In the experiments, only supervised pre-training is considered. Since self-supervised representation learning has recently become a popular way to pre-train the model, it would be great if it is included in the comparison to reveal the best strategy of pre-training in double-transfer setting.
>
> We thank the Reviewer for their thoughtful comment. We agree on the importance of self-supervised representation learning and consider it a direction worth pursuing for future work.
>
> We decided to focus our current analysis to classification-based pre-training for manuscript space limitations, target audience, and computational constraints: firstly, we narrowed down the focus of the work on the relevance of the backbone, fine-tuning, and normalization layer choices. Secondly, we considered the availability of self-pretrained models online and observed that a few self-supervised models are shared (and mostly small ResNet50) making them less accessible to the community. As it can be noticed from Table 2, improving the backbone (from ResNet50 to ConvNext-B) yields a 10% accuracy gain. We do not expect that self-supervision alone would bridge the gap. Since we aim to provide helpful insights for practitioners facing SF-UDA challenges in realistic scenarios, we decided to limit the investigation to the role of the architecture.
>
> We remark that Tables 8 and 9 in Appendix F provide preliminary results on the effect of a different pre-training strategy, _miil_ (Ridnik et al., 2021).
>
>
> > Does "DGen" mean evaluation of a target model under the domain shift setting? This terminology is quite confusing. (it seems to indicate a certain domain generalization method.)
>
> Thank you for pointing this out. Yes, in our work we considered “DGen” as the evaluation of a model under a domain shift setting (without any adaptation), but we agree that it was not made explicit: we updated the manuscript to improve the clarity on the meaning of “DGen” (see Section 2, Domain Generalization).
>
>
> > Why does "Generalization LP" perform best in Fig. 1? Is it showing the accuracy at the source domain? If so, this is also confusing, because the other subfigures show the accuracy at the target domain.
>
> When we refer to Generalization we refer to the standard setting where the training and test sets are sampled from the same domain distribution, while DGen and SF-UDA always consider a pair of domains and the performance is evaluated on the target one. We updated the caption of Fig.1 to make this distinction explicit and more clear.

---

> > ### Comment · Reviewer_cjNK · 2022-11-17
> > **Thanks for the reply**
> >
> > I would like to appreciate the authors for making some clarification related to my concerns.
> >
> > At this moment, I keep my score due to the following two reasons (mainly the first one):
> > - As also pointed out by ReviewerR8P1 and GrZy, the findings through the experiments are not so appealing. I agree with that the authors "extensively demonstrate those findings in a different context, in which we have no guarantees." However, if these findings are not much different from the existing related studies, it is not clear to see the importance of specifically considering this context, which is double-transfer setting, for future ML research.
> > - The confusing notations remain. According to the authors' explanation, "DGen -> X" means a specific setting of  "Task -> Method," which is not intuitively understandable.

---

> > > ### Author Response · Authors · 2022-11-18
> > > **Answers to New Comments. Part 1 of 3.**
> > >
> > > We now proceed with answering the new comments by Reviewer CjNk’s.
> > >
> > > > As also pointed out by ReviewerR8P1 and GrZy, the findings through the experiments are not so appealing.
> > >
> > > Please consider that both Reviewers R8P1 and GrZy also pointed out several substantial strengths of our paper and its potential impact. The interactive discussion is currently open, and we are pleased to engage in the discussion and answer all the questions and doubts put forward by the Reviewers to clarify the value of our contributions. We point Reviewer CjNk to our detailed responses to Reviewers R8P1 and GrZy, and invite Reviewer CjNk to also take those answers into account for evaluating the novelty and impact of our findings.

---

> > > ### Author Response · Authors · 2022-11-18
> > > **Answers to New Comments. Part 2 of 3.**
> > >
> > > > I agree with that the authors "extensively demonstrate those findings in a different context, in which we have no guarantees." However, if these findings are not much different from the existing related studies,
> > >
> > > We would like to further clarify that in our work we present several relevant findings, all significant and useful in their own right and backed by extensive and rigorous experimental analysis. Also note that SF-UDA is significantly different from the settings of the existing studies the Reviewer is referring to.
> > >
> > > Importantly, our findings belong to *three different categories in terms of novelty*:
> > >
> > > 1) **Completely novel findings:** They are not present in the literature and thus provide *new and valuable understanding* of the challenging SF-UDA learning problem and the associated learning method design choices:
> > >     - ***BN can be brittle, LN proves more robust in SF-UDA:*** Previous works showed the importance of BN Layers for UDA and some of them built ad-hoc techniques to use BN Layers while enabling transfer (Li et al., 2017). Differently, we show that architectures with BN are very sensitive to domain shift only if fine-tuning is performed, i.e., if BN statistics are updated to the source ones.
> > >  Furthermore, we show that techniques like AdaBN *can easily fail* in these cases. We demonstrate *for the first time* that Layer Normalization, on the other hand, is less prone to failures (Tab. 4) and achieves improved performances (Tab.5) in SF-UDA.
> > >     - ***Double-transfer is relevant in many real scenarios:*** Our work is the very first to analyze the effect of design choices on the performances of *both transfers* taken separately and jointly.
> > >     - ***SF-UDA success rate analysis:*** Understanding how likely is a method to fail is of the utmost relevance for practitioners. In most of the current literature, only the successful results (on the best datasets) are presented, as pointed out by (Kim et al., 2022).
> > > Instead, we extensively evaluate *realistic failure rates* on a large number of domain shifts and large set of architectures.
> > >     - ***Best practices for practitioners:*** The findings of this work provide a set of cally-based rules and key factors to consider while tackling the SF-UDA problem, thus helping to support practitioners in bridging the domain shift in real-world scenarios.
> > > 2) **Findings that challenge the literature and/or common practice:** They come to *different conclusions* from those observed in the literature or commonly applied by practitioners. Therefore, these are *very useful in practice*. Moreover, they are also valuable as they correct/fill gaps in the literature and point researchers towards understudied phenomena occurring in the double-transfer setting:
> > >     - ***Fine-tuning on the source domain can degrade adaptation performance:*** It is common practice in DA to fine-tune the model on the source domain. We show that the overlooked fine-tuning step on the source *can actually be detrimental* when there is a large shift in the normalization statistics.
> > > This is especially evident with BN and DA strategies based on a shared feature extractor. To a smaller extent, we reach similar conclusions for LN as well. Algorithms adapting the feature extractor to the target, e.g., SHOT, prove more robust.
> > > 3) **Findings that *extend* previous ones from other contexts to the SF-UDA problem setting.** Common practices and design choices may not necessarily apply to SF-UDA, which is a more constrained setting:
> > >     - ***Relevance of the backbone choice; top-1 performance on IN (known for UDA)***
> > > (Kornblith et al., 2020) showed how the relevance of IN1k accuracy highly correlates with UDA performances. We *extend* the results to SF-UDA to several more recent and appealing architectures (ConvNext, Transformers, etc.), and to asignificantly larger number of datasets. Furthermore, we indicate and quantify that knowledge about backbone pre-training aids model selection. Despite similar IN1k accuracy, models pre-trained on IN21k perform best.
> > >     - ***Pretraining on IN21k helps ViT training:*** (Dosovitskiy et al. 2021) demonstrates that a larger pretraining on IN21k or JFT-300M improves the performance of ViT on a downstream task. The setting considers the *first transfer only*, i.e., there is *no adaptation* to the target. Also, no results on other architectures are provided.
> > > In contrast, we demonstrate the gains in a DA setting where we can leverage source knowledge to *adapt to the target*. Moreover, we provide results on the benefit of a larger pre-training on *different families of architectures*, e.g., VGG, ResNet, ConvNext, etc.

---

> > > ### Author Response · Authors · 2022-11-18
> > > **Answers to New Comments. Part 3 of 3.**
> > >
> > > > it is not clear to see the importance of specifically considering this context, which is double-transfer setting, for future ML research.
> > >
> > > The presented results quantitatively demonstrate the *large impact* that the key design choices we identified have for SF-UDA:
> > > 1) In Table 2, we show the relevance of the backbone. Consider SHOT+FT-IN21k, moving from the benchmark ResNet50-W to a ConvNext B yields an **8.7% accuracy gain**. Similar results are obtained for SCA, that achieves a **10.3% improvement**.
> > > 2) Table 2 also shows evidence on the effectiveness of a larger pre-training: both SCA and FT+SHOT benefit from IN21k. The accuracy **improvement is up to ~10%** for the large ConvNext B. A lower failure rate is observed in Table 4 where IN is **5% more likely to fail.**
> > > 3) Fine-tuning on the source can be detrimental for adaptation, especially when BN is employed. Special attention is needed. In Table 3 we can observe that an algorithm as SCA that shares the same feature extractor between source and target suffers a **performance drop of -4.6% with BN** for the IN21k setting. In contrast, **LN models benefit from fine-tuning** on average.
> > > 4) LN proves more robust than BN. Evidence is provided in table 4, where in all settings it has a lower failure rate. **The drop is critical for FT+SCA, namely -26%.**
> > >
> > > Clarity
> > > ------
> > >
> > > > The confusing notations remain. According to the authors' explanation, "DGen -> X" means a specific setting of "Task -> Method," which is not intuitively understandable.
> > >
> > > We updated Sections 3.1, 4.3, and 4.4 and Figures 2 and 3 (both subtitles and captions) to remove ambiguities and improve clarity.
> > >
> > > Conclusions
> > > --------
> > >
> > > We believe that the argument that other works have found similar results in other settings is too generic and does not accurately characterize our work. In this response, we further provided a clear demonstration of the usefulness of our results, in their rigor, and in the future possible research directions (many of them suggested by the Reviewers) that our work opens to the ICLR research community, besides its value for practitioners.
> > >
> > > To summarize, our setup and experiments are rigorous and our findings are manifold, impactful, and well-specified. Taken individually, they may or may not appear surprising depending on the individual reader's background. However, we strongly believe their novelty and impact are solid, very useful for practitioners and researchers and, as a consequence, that this work would be a really valuable addition to the previous literature.
> > >
> > >
> > >
> > > **References**
> > >
> > > Dosovitskiy, Alexey, et al. "An Image is Worth 16x16 Words: Transformers for Image Recognition at Scale." International Conference on Learning Representations. 2020.
> > >
> > > Yanghao Li, Naiyan Wang, Jianping Shi, Jiaying Liu, and Xiaodi Hou. Revisiting batch normalization for practical domain adaptation. In International Conference on Learning Representations, 2017.
> > >
> > > Tal Ridnik, Emanuel Ben-Baruch, Asaf Noy, and Lihi Zelnik-Manor. Imagenet-21k pre-training for the masses. In Proceedings of the Neural Information Processing Systems Track on Datasets and Benchmarks, 2021.
> > >
> > > Donghyun Kim, Kaihong Wang, Stan Sclaroff, and Kate Saenko. A broad study of pre-training for domain generalization and adaptation.
> > >  arXiv preprint arXiv:2203.11819, 2022.
> > >
> > > Simon Kornblith, Jonathon Shlens, and Quoc V Le. Do better imagenet models transfer better? In Proceedings of the IEEE/CVF Conference on Computer Vision and Pattern Recognition, pp. 2661–2671, 2019.

---

> ### Author Response · Authors · 2022-11-14
> **Answers to Reviewer cjNK comments. Part 2 of 2.**
>
> > "DGen -> X" in Fig. 2 is also confusing. It sounds like training the model with a certain domain generalization method at the source domain and then conducting X to adapt the model to the target domain (but it does not mean such a thing).
>
> We think that with the modifications applied to the manuscript and the explicit remark on the meaning of “DGen” used across the paper, also Fig. 2 should become more clear as a consequence. Nevertheless, we updated the caption of Fig. 2 to clarify the notation.
>
> Based on the responses to the feedback and the released [Code and full Results](https://drive.google.com/file/d/14cXcFPCuS4dmD5y3I-dJNH94ETlVSxPc/view?usp=sharing), we hope the Reviewer will consider increasing the score for our paper.
> If needed, we are glad to provide further clarifications.
>
> **References**
>
> Dosovitskiy, Alexey, et al. "An Image is Worth 16x16 Words: Transformers for Image Recognition at Scale." International Conference on Learning Representations. 2020.
>
> Yanghao Li, Naiyan Wang, Jianping Shi, Jiaying Liu, and Xiaodi Hou. Revisiting batch normalization for practical domain adaptation. In International Conference on Learning Representations, 2017.
>
> Tal Ridnik, Emanuel Ben-Baruch, Asaf Noy, and Lihi Zelnik-Manor. Imagenet-21k pre-training for the masses. In Proceedings of the Neural Information Processing Systems Track on Datasets and Benchmarks, 2021.

---

> ### Author Response · Authors · 2022-11-18
> **Added new experiments: Self-supervised pre-training in SF-UDA**
>
> We are again thankful to Reviewer CjNk for their suggestion to also consider the self-supervised pre-training setting in our experiments, due to its widespread use and relevance to the field.
>
> We decided to follow the Reviewer’s suggestion by carrying out new experiments using three self-supervised pre-training methods to analyze their performance in the double-transfer SF-UDA setting. We are glad to point the interested reader to the full results and detailed discussion in Section 4.2 and Appendix J (Tables 14 and 15).
>
> In a nutshell, we present additional experiments to compare supervised pre-training with Self-Supervised Learning (SSL) strategies (DINO, MOCO v1, and MOCO v2) for pre-training and we show that, despite the promising results, there is still a gap in performance between the two approaches.

---

### Official Review · Reviewer_1ukJ · 2022-10-25

**Confidence:** 5
**Correctness:** 3
**Technical Novelty And Significance:** 1
**Empirical Novelty And Significance:** 2
**Recommendation:** 5

**Clarity, Quality, Novelty And Reproducibility:**

The description of new algorithm is not clear, and the writing and paper organization should be polished.

**Strength And Weaknesses:**

This work explored source-free domain adaptation setting (SF-UDA). Different from conventional UDA scenario, SF-UDA assumes that model adaptation of target domain only accesses well-trained source model without well-labeled source data. Overall, in my opinion, this work is only an experimental report instead of a complete research paper due to two main reasons.

1. In Section “Method”, the current manuscript only provided many details of the existing SF-UDA work (SHOT) without description of new proposed algorithm. Thus, this current version lacks a complete novel algorithm.

2. Currently, there are a lot of explorations on SF-UDA topic [1, 2, 3]. However, the current work does not make fair comparisons with them. Thus, the experiments are not enough.

[1] Chu, Tong, et al. "Denoised maximum classifier discrepancy for source free unsupervised domain adaptation." Thirty-Sixth AAAI Conference on Artificial Intelligence (AAAI-22). Vol. 2. 2022.

[2] Ding, Ning, et al. "Source-Free Domain Adaptation via Distribution Estimation." Proceedings of the IEEE/CVF Conference on Computer Vision and Pattern Recognition. 2022.

[3] Kundu, Jogendra Nath, et al. "Balancing discriminability and transferability for source-free domain adaptation." International Conference on Machine Learning. PMLR, 2022.


**Summary Of The Paper:**

This work explored source-free domain adaptation setting (SF-UDA). Different from conventional UDA scenario, SF-UDA assumes that model adaptation of target domain only accesses well-trained source model without well-labeled source data.

**Summary Of The Review:**

This work explored source-free domain adaptation setting (SF-UDA). Different from conventional UDA scenario, SF-UDA assumes that model adaptation of target domain only accesses well-trained source model without well-labeled source data. Overall, in my opinion, this work is only an experimental report instead of a complete research paper due to two main reasons.

1. In Section “Method”, the current manuscript only provided many details of the existing SF-UDA work (SHOT) without description of new proposed algorithm. Thus, this current version lacks a complete novel algorithm.

2. Currently, there are a lot of explorations on SF-UDA topic [1, 2, 3]. However, the current work does not make fair comparisons with them. Thus, the experiments are not enough.

[1] Chu, Tong, et al. "Denoised maximum classifier discrepancy for source free unsupervised domain adaptation." Thirty-Sixth AAAI Conference on Artificial Intelligence (AAAI-22). Vol. 2. 2022.

[2] Ding, Ning, et al. "Source-Free Domain Adaptation via Distribution Estimation." Proceedings of the IEEE/CVF Conference on Computer Vision and Pattern Recognition. 2022.

[3] Kundu, Jogendra Nath, et al. "Balancing discriminability and transferability for source-free domain adaptation." International Conference on Machine Learning. PMLR, 2022.

---

> ### Author Response · Authors · 2022-11-14
> **Answers to Reviewer 1ukJ comments.**
>
> > In Section “Method”, the current manuscript only provided many details of the existing SF-UDA work (SHOT) without description of new proposed algorithm. Thus, this current version lacks a complete novel algorithm.
>
> We thank the Reviewer for their comment. We respectfully disagree with their implied expectation that every research paper should necessarily describe a new proposed algorithm. Every research paper should provide new, relevant, impactful knowledge (including empirical, theoretical, for practitioners, etc. (ICLR PCs, 2023)).
>
> We remark that this work aims to guide practitioners in the key design choices underlying SF-UDA methods, providing extensive and consistent empirical evidence on backbones, fine-tuning, and normalization. On these lines, our consistent findings show to which degree often-overlooked design choices might impact downstream results. In Table 2, switching from ResNet50 to ConvNext-B boosts the performance by ~10% in all considered settings; on average, fine-tuning on SHOT results in a 2.7% gain (Tab. 3, 1st line). Finally, fine-tuning is detrimental when batch normalization is applied and source statistics are far from the target ones. Moreover, Table 5 highlights how SF-UDA algorithms are competitive with UDA solutions despite the lack of source data at adaptation stage.
>
> To sum up, we believe our contributions convey strong and impactful value to the field. Follow-up work planned for the future will be aimed at shedding light on DA algorithms that have been shown (Kim, Donghyun, et al., 2022) to be prone to failure beyond possibly overfit benchmarks, i.e., introduced as novel methods, yet often too brittle when rigorously evaluated in more realistic scenarios.
>
>
> > Currently, there are a lot of explorations on SF-UDA topic [1, 2, 3]. However, the current work does not make fair comparisons with them. Thus, the experiments are not enough.
>
> We acknowledge there needs to be a better understanding regarding the scope of our work. We are not interested in benchmarking and comparing all SF-UDA methods. On the contrary, our objective is to show the setting's potential and the realistic results outside standard benchmarks. Experiments systematically reveal that the performance of different SF-UDA methods varies very significantly depending on the employed architectures, pre-trainings, normalization strategies, and datasets. We challenge common literature that often fail to address the limitations of their approaches. There is no best performing method, e.g., on average, FT+SHOT outperforms SCA, however there are cases in which SCA achieves better results. Nevertheless, we demonstrate the benefits of LayerNorm with respect to BatchNorm (Tab. 4) and support pre-training on the larger IN21K (Tab. 2).
>
> We stand with our experimental setup being rigorous and precise: an extensive and fair evaluation of design choices for the applicability of SF-UDA approaches. Our large-scale experiments consistently support the paper’s claims. We are confident that the proposed analysis is a solid contribution to SF-UDA research and would be beneficial for researchers and practitioners in the field.
>
> **References**
>
> ICLR23 Program Chairs, ICLR 2023 Reviewer Guide, Reviewing a submission: step-by-step, accessed 10 November 2022, [https://iclr.cc/Conferences/2023/ReviewerGuide](https://iclr.cc/Conferences/2023/ReviewerGuide).
>
> Kim, Donghyun, et al. "A Broad Study of Pre-training for Domain Generalization and Adaptation." arXiv preprint arXiv:2203.11819 (2022).

---

### Official Review · Reviewer_GrZy · 2022-10-31

**Confidence:** 4
**Correctness:** 4
**Technical Novelty And Significance:** 2
**Empirical Novelty And Significance:** 2
**Recommendation:** 5

**Clarity, Quality, Novelty And Reproducibility:**

+ The paper is well organized and their proposed approach is generally well described.
+ This study seeks to clarify the impact of the main design choices in SF-UDA approaches, and investigate the strengths and failure modes of SF-UDA methods. This paper is a details empirical study with mostly pre-existing existing methods. The study approach is well-motivated, but as expected, there is limited conceptual innovation.
+ The code was not made available, although the methods are well described. Since the experimental data and code are not available (released upon acceptance), there is a concern that the results in this paper would be difficult for a reader to reproduce.



**Details Of Ethics Concerns:**

None.

**Strength And Weaknesses:**

Strengths:
+ The authors provide an in-depth empirical analysis of design choices for SF-UDA with several UDA and SF-UDA methods, backbone architectures, and datasets.  This SF-UDA  setting is particularly challenging and still remains understudied. I anticipate that this study and its results should be useful to researchers and practitioners working in this area.
+ The methods and experimental methodology are described in enough detail to understand the paper.
+ The conclusions of this study are interesting. For instance, it is useful to know that State-of-art SF-UDA methods, like SHOT, are competitive with state-of-the-art UDA methods, but can be implemented much more efficiently.
+ The supplementary material provides additional algorithmic and implementation details on baseline methods, and experimental results that help support the paper.

Weaknesses:
- Some definitions are not very clear at the beginning: DA, DG, etc., and how they relate to SOA methods for multi-task and multi-domain learning.
- The authors do not provide a critical analysis of methods, or highlight from the outset the key challenges facing practitioners or researchers working on SF-UDA methods and problems.
- The empirical results are not fully convincing. The scope of this study is limited to standard image classification problems, using common datasets. There is no results or discussion on the results that would be obtained from real-world data captured in the wild.  The authors should explore the impact on the performance of, e.g., class imbalance, noisy image data, etc.
- The authors should measure domain shift empirically in order to characterize the difficulty of an SF-UDA problem.
- Results on FT+NRC should not be included in the results if it is trained using only trained on 15% of the DomainNet data.
- Finally, the authors should asses and compare the computational complexity of different SF-UDA methods, and show the differences w.r.t. UDA methods.


**Summary Of The Paper:**

This paper is about Source-free Unsupervised Domain Adaptation (SF-UDA).  The authors provide an analysis of the impact of main design choices in SF-UDA through a large-scale empirical study with several CNN and ViT models, methods, and datasets for image classification. In particular, they observe the normalization approach, pretraining strategy, and backbone architecture are the most impactful design choices, and propose best practices for robust SF-UDA methods. The authors also seek to study the strengths and failure modes of SF-UDA methods and compare them with UDA approaches. Empirical results show that SF-UDA methods like SHOT can provide accuracy comparable with that of state-of-the-art UDA methods (at much lower data and computational cost, and better than standard benchmarks and backbone architectures.

**Summary Of The Review:**

Overall this is good quality submission.  The proposed study is well-motivated, although the experimental validation could be improved.

---

> ### Author Response · Authors · 2022-11-14
> **Answers to Reviewer GrZy comments. Part 1 of 3.**
>
>
> > The authors provide an in-depth empirical analysis of design choices for SF-UDA with several UDA and SF-UDA methods, backbone architectures, and datasets. This SF-UDA setting is particularly challenging and still remains understudied. I anticipate that this study and its results should be useful to researchers and practitioners working in this area.
>
> We thank the Reviewer for highlighting the core contributions of our work. A deeper and empirically grounded understanding of key design choices across individual SF-UDA methods indeed represents a step forward for both practitioners and researchers to effectively apply and design new methods in this challenging and impactful scenario.
>
>
> > The methods and experimental methodology are described in enough detail to understand the paper.
>
> We tried to structure the paper as clearly as possible for it to be accessible to both researchers and practitioners.
>
>
> > The conclusions of this study are interesting. For instance, it is useful to know that State-of-art SF-UDA methods, like SHOT, are competitive with state-of-the-art UDA methods, but can be implemented much more efficiently.
>
> We acknowledge the Reviewer’s comment on the usefulness and novelty of our results.
>
>
> > The supplementary material provides additional algorithmic and implementation details on baseline methods, and experimental results that help support the paper.
>
> We appreciate that the Reviewer found the supplementary material useful.
> To further support our paper’s contributions, we decided to anonymously release the full results and experimental code to the Reviewers. The additional material is now available at [Code](https://drive.google.com/file/d/14cXcFPCuS4dmD5y3I-dJNH94ETlVSxPc/view?usp=sharing)
>
>
> > Some definitions are not very clear at the beginning: DA, DG, etc., and how they relate to SOA methods for multi-task and multi-domain learning.
>
> We thank the Reviewer for the interesting connection and feedback. We acknowledge that the transferability of the learned representations is pivotal in many scenarios and recognize that the introduction of more general settings such as MDL (Joshi et al.,  2012) and MTL (Caruana, 1997) would provide a more comprehensive overview of the current state of art in transfer learning.
>
> We remark that our work focuses on providing an experimental analysis of the key components of Source-free UDA methods, also identifying best practices of interest to practitioners who may be less familiar with the broader literature. Hence, striving for clarity and conciseness, we focused the discussion on the essential concepts to understand the SF-UDA setting and our findings. Specifically, we decided to introduce DGen in equation (1) as a transfer baseline and its differences with the classic UDA setting. Then, we defined SF-UDA in equations (2-3) and contrasted it to the previous settings. We restricted ourselves to examining the single-source and single-target domain setting, since it is the most common variant, and to avoid introducing additional confounding factors. For the above reasons, we opted for narrowing down the definitions to the minimal concepts on which the work is built.
>
> However, we agree with the Reviewer's comment on the particular relevance of MTL and MDL, and we decided to extend Section 1 of the manuscript mentioning the suggested settings and pointing the reader to relevant works on these topics. Furthermore, we updated the manuscript to introduce DGen and relate it to UDA, and SF-UDA in the introduction section.
>
>
> > The authors do not provide a critical analysis of methods, or highlight from the outset the key challenges facing practitioners or researchers working on SF-UDA methods and problems.
>
> While it is true that we did not provide a complete overview of all the challenges that practitioners or researchers that work on SF-UDA have to face (also due to space limitations), our work is fully focused on the key challenges that we consider to be the most important and/or understudied. In particular, in practical applications it is crucial to decide on which architecture to use, how to pre-train it, whether it is useful or not to fine-tune it on the source domain, and how likely is the SF-UDA method to succeed. At the end of the Introduction we summarized the answers to these questions and we analyzed them in great detail throughout the work.
>
> Furthermore, we think that our key findings can also help researchers to achieve a better understanding of double-transfer and potentially promote further studies including different pre-trainings, SF-UDA methods and tasks beyond Image Classification.

---

> ### Author Response · Authors · 2022-11-14
> **Answers to Reviewer GrZy comments. Part 2 of 3.**
>
> > The empirical results are not fully convincing. The scope of this study is limited to standard image classification problems, using common datasets. There is no results or discussion on the results that would be obtained from real-world data captured in the wild. The authors should explore the impact on the performance of, e.g., class imbalance, noisy image data, etc.
>
> Thank you for the invaluable suggestions. We agree that studying SF-UDA on real-world data with noise on images/labels, class imbalance, and tasks beyond image classification would be really helpful towards a broader understanding of the strengths and weaknesses of SF-UDA methods “in the wild”. Our work represents a significant and necessary first step enabling further investigations in these directions. We limit the range of experiments to image classification to provide sound and reliable results while controlling for several key design choices. We are convinced that extending our rigorous empirical analysis of SF-UDA methods to more complex tasks beyond image classification would be an interesting and impactful research direction.
>
> One of the main motivations of our work emerges from the critical problems in the evaluation of UDA algorithms for Image Classification. The UDA validation protocol relies on standard benchmarks, i.e., usually two datasets chosen between Office31, VISDA and Office-Home, and it is limited to a few common architectures, e.g., ResNet50/101. It is important to note that, as recently reported by (Kim et al., 2022), changing the architecture or the dataset leads to a drop in performance of several UDA methods. We encountered many difficulties in the reproduction of UDA methods on the same benchmarks presented in the official papers and/or the applications of such methods on different architectures/datasets.
> Such observations on the reproducibility limitations of UDA contributed to our motivation to conduct this study on the relatively newer SF-UDA methods.
>
> Our findings show that SF-UDA techniques consistently benefit adaptation, yielding performance gains with respect to the DGen baselines across a wide range of datasets and architectures. In particular, we consider a large number of architectures and backbones to provide an empirically sound analysis of what can happen when applying SF-UDA in practice depending on key design choices. For a fair comparison, we keep the hyperparameters fixed and evaluate the performance/robustness of the considered methods. In our view, the common practice of benchmarking a method on a single architecture and a single public dataset provides inadequate information on its performance in real-world scenarios. We cover this gap by performing a large-scale evaluation (74 different domain shifts, many of them from datasets beyond those commonly employed in DA benchmarks).
>
> It is also worth noting that the datasets employed in our experiments are quite diverse (different types of domains, number of images, number of classes, etc.). Some of them even present large imbalances in the number of samples for each domain (e.g., in Modern-Office31 there is a domain with 498 images and another one with 3100; in DomainNet a domain with 48833 images and another on with 175327, etc.).
>
> Hence, our results would be informative for estimating the performance/reliability of SF-UDA methods and informing design choices in real-world scenarios. It is our strong belief that this work would contribute to the literature and inspire future research in SF-UDA.
>
>
> > The authors should measure domain shift empirically in order to characterize the difficulty of an SF-UDA problem.
>
> Measuring domain shift in absolute terms is still a challenging open research problem which goes beyond the scope of this work. We follow standard practices in the DA literature without explicitly measuring the shift. Still, note that DGen itself implicitly provides an estimate of the distance between two domains as the drop in accuracy when moving from the training domain to the target domain. We refer the Reviewer to another line of research (Zamir et al., 2018; Achille et al., 2019) focused on probing the distance among domains.
>
> Instead, in our experimental analysis we aim to show how SF-UDA methods may perform in a practical scenario by analyzing many combinations of source-target domain pairs with a broad range of shifts.

---

> ### Author Response · Authors · 2022-11-14
> **Answers to Reviewer GrZy comments. Part 3 of 3**
>
> > Results on FT+NRC should not be included in the results if it is trained using only trained on 15% of the DomainNet data.
>
> We acknowledge that the experimental setting in Section 4.5 and Table 5 is not fully representative of the potential predictive performances of the FT+NRC SF-UDA method. However, we respectfully disagree on removing this specific experiment. In fact, the lightweight setting we employed (15% data subsampling) is sufficient to support our claims. Table 5 aims at demonstrating the competitive performances of SF-UDA with respect to  UDA methods. To this end, among other experiments, we compare FT+NRC (an SF-UDA method) trained and evaluated on a fraction of the available DomainNet data (15%) to classical UDA algorithms trained and evaluated on the entire dataset (100%). Crucially, despite the limited training data, FT+NRC still compares favorably and sometimes outperforms UDA methods. A larger training set would likely further improve the performance of FT+NRC, making it even more competitive with UDA methods, thus strengthening our claim. Note that the large size of DomainNet allows us to obtain a good estimate of the test performances of the method FT+NRC even when subsampled. To give an idea, 15% of the smallest available domain in DomainNet counts 7324 samples, which is larger than the full Modern Office31 dataset with all its 4 domains.
>
> We are thankful to the Reviewer for pointing this out and proceeded to clarify this point in the revised version of the paper (see Section 4.5).
>
> > Finally, the authors should asses and compare the computational complexity of different SF-UDA methods, and show the differences w.r.t. UDA methods.
>
> We considered introducing a table with the computational complexities of the different algorithms for comparison. However, from our experience, those provide little information on the actual real-world time complexity, which heavily depends on the specific dataset, optimizer, model architecture, implementation, framework, hardware, and combinations thereof.
>
> For this reason, we found it more informative to report empirically-measured reference training times in Appendix H (see Table 12). We would like to underscore that the adaptation algorithm itself does not make the main difference (in terms of computations) between UDA and SF-UDA. The computational efficiency rather arises from the possibility of splitting SF-UDA methods into two separate phases. In practical scenarios, we can train on the source domain beforehand while performing the adaptation when the target data becomes available (for instance, it might be performed on more resource-constrained devices). This is impossible for classical UDA methods that need a single training with both the source and the target data available at the same time.
>
> In light of these clarifications, would the Reviewer consider increasing the score for our paper? If not, please let us know any additional changes or clarifications that would improve our work.
>
> **References**
>
> Achille, Alessandro, et al. "Task2vec: Task embedding for meta-learning." Proceedings of the IEEE/CVF international conference on computer vision. 2019.
>
> Caruana, Rich. "Multitask learning." Machine learning 28.1 (1997): 41-75.
>
> Joshi, Mahesh, et al. "Multi-domain learning: when do domains matter?." Proceedings of the 2012 Joint Conference on Empirical Methods in Natural Language Processing and Computational Natural Language Learning. 2012.
>
> Kim, Donghyun, et al. "A Broad Study of Pre-training for Domain Generalization and Adaptation." arXiv preprint arXiv:2203.11819 (2022).
>
> Zamir, Amir R., et al. "Taskonomy: Disentangling task transfer learning." Proceedings of the IEEE conference on computer vision and pattern recognition. 2018.

---

> ### Author Response · Authors · 2022-11-18
> **Added New Computational Time Experiments for UDA**
>
> > Finally, the authors should asses and compare the computational complexity of different SF-UDA methods, and show the differences w.r.t. UDA methods.
>
> Thank you for the suggestion. We added in Appendix H the computational times of the UDA method DANN (Ganin et al.) to have a reference comparison with SCA and SHOT. We did the explicit choice of reporting the times of DANN because it is representative of adversarial approaches and it is foundational for recent state-of-the-art methods, e.g., FixBi (Na et al., see issues in the [official repository](https://github.com/NaJaeMin92/FixBi)).
>
> Despite the shorter time needed for a single epoch compared to SHOT, DANN requires a longer training (usually ~100 epochs), while SHOT usually needs ~15 epochs only. Note that SCA proves faster by a large margin.
>
>
> **References**
>
> - Ganin, Yaroslav, et al. "Domain-adversarial training of neural networks." The journal of machine learning research 17.1 (2016): 2096-2030.
> - Long, Mingsheng, et al. "Conditional adversarial domain adaptation." Advances in neural information processing systems 31 (2018).
> - Na, Jaemin, et al. "Fixbi: Bridging domain spaces for unsupervised domain adaptation." Proceedings of the IEEE/CVF Conference on Computer Vision and Pattern Recognition. 2021.

---

### Author Response · Authors · 2022-11-14
**General Comment to Reviewers and Area Chair**

We would like to thank the Area Chair for chairing our submission and all the Reviewers for their constructive feedback and valuable comments, which allowed us to considerably improve the manuscript. We acknowledge that the Reviewers recognize the relevance of the challenging learning setting investigated in this work. The evaluation of key design elements for Source-Free Unsupervised Domain Adaptation (SF-UDA) methods is indeed still understudied and valuable for practitioners and researchers in this growing field.

We are encouraged by Reviewers GrZy, cjNK, and R8P1’s consensus on the benefits of large-scale experiments and on the significant value of the results presented in the paper.

We proceed by addressing the concerns raised by each Reviewer in individual comments and uploading a revised version of the manuscript, where the green color highlights our changes.

Also note that, following Reviewers comments, we anonymously released the experimental code and full results, which are available for download at: [Code](https://drive.google.com/file/d/14cXcFPCuS4dmD5y3I-dJNH94ETlVSxPc/view?usp=sharing)

---

> ### Author Response · Authors · 2022-11-18
> **New Experiments: Self-supervised Pre-training in SF-UDA**
>
> Due to its widespread use and relevance to the field, we followed the insightful suggestion of Reviewer CjNk to also consider the self-supervised pre-training setting in our experiments. In particular, we carried out new experiments using three self-supervised pre-training methods to analyze their performance in the double-transfer SF-UDA setting. We are glad to point the interested reader to the full results and detailed discussion in Section 4.2 and Appendix J (Tables 14 and 15).
>
> In a nutshell, we present additional experiments to compare supervised pre-training with Self-Supervised Learning (SSL) strategies (DINO, MOCO v1, and MOCO v2) for pre-training and we show that, despite the promising results, there is still a gap in performance between the two approaches.

---

### Author Response · Authors · 2022-12-07
**Summary of Improvements and Responses**

Dear all,

As the end of the interactive discussion period is approaching, **we would like to summarize the significant improvements to our submission** made possible by the Reviewers' in-depth feedback.

We extensively addressed all the concerns and doubts raised by the Reviewers. Updates to the manuscript are colored in green in the latest version of the PDF.

In particular:
1. **We clarified novelty** [[link]](https://openreview.net/forum?id=-PL1Gk4jt7&noteId=dhZ4t8umhHa) **and impact** [[link]](https://openreview.net/forum?id=-PL1Gk4jt7&noteId=zAIZWC0QpVE). Such responses might be of interest to all the Reviewers and the AC.
2. We carried out **new experiments on self-supervised pre-training**, as suggested by Rev. cjNK. Results are reported in Section 4.2 and Appendix J (Tables 14 and 15). More details available [[here]](https://openreview.net/forum?id=-PL1Gk4jt7&noteId=gd7I2ZMxMP).
3. **We added the computational times of the UDA method DANN** (Ganin et al., 2016) in Appendix H to have a reference comparison with SCA and SHOT, as suggested by Reviewer GrZy.
4. **We improved the clarity** of the manuscript. Following Reviewer cjNK's suggestion, we stressed the meaning of the employed notation and changed the subtitles in Fig. 2 accordingly. As suggested by Reviewer GrZy, we improved the definitions of DA and DG, and we related them to state-of-the-art methods.
5. Encouraged by multiple Reviewers, **we released the anonymized code and full results**: [[download]](https://drive.google.com/file/d/14cXcFPCuS4dmD5y3I-dJNH94ETlVSxPc/view?usp=sharing).

For these reasons, we believe that our submission has been significantly strengthened and its multiple impactful contributions exhaustively highlighted. This work represents a significant and timely step forward in shedding light on SF-UDA methods.

**We are looking forward to addressing any outstanding questions** and we trust that our submission will be reassessed based on our improvements and responses.

Thank you for your time.

Best regards,

The authors.

---

### Decision · Program_Chairs · 2023-01-20

**Decision:**

Reject

**Justification For Why Not Higher Score:**

- Empirical study is limited to 2 methods, which raises questions about its generalizability
- Findings are not too surprising (e.g. architecture affects performance) and partly alluded to in the literature

**Justification For Why Not Lower Score:**

N/A

**Metareview: Summary, Strengths And Weaknesses:**

This paper studies methods for source-free unsupervised domain adaptation focusing on the "double-transfer" setting and performs an empirical study on the factors affecting their performance. While reviewers appreciated that the experiments were extensive in terms of datasets, domain pairs and models evaluated, they had unresolved concerns that only 2 adaptation methods were evaluated (even after the authors' response), thus limiting the generality of the conclusions. They also had concerns that many of the findings were not surprising, partly as similar results have appeared in the literature. As such, the paper does not currently meet the bar for ICLR.

That said, the study has promise and can likely be publishable with further work. Beyond expanding the set of methods included in the study, it would be useful to discuss the settings under which the identified factors (e.g. architecture, pre-training strategy) can actually be controlled; in some applications it would not be possible to control the source-model being provided, in which case a greater focus on the factors affecting adaptation would be more relevant as alluded to by the reviewers. It could also be helpful to take reference from a related study in the NLP literature -- "A Comparison of Strategies for Source-Free Domain Adaptation", ACL 2022.